# Experimental Studies on TiO_2_ NT with Metal Dopants through Co-Precipitation, Sol–Gel, Hydrothermal Scheme and Corresponding Computational Molecular Evaluations

**DOI:** 10.3390/ma16083076

**Published:** 2023-04-13

**Authors:** Eduardo Patricio Estévez Ruiz, Joaquín López Lago, Saravana Prakash Thirumuruganandham

**Affiliations:** 1Centro de Investigación de Ciencias Humanas y de la Educación (CICHE), Universidad Indoamérica, Ambato 180103, Ecuador; 2Grupo de Polímeros, Departamento de Física y Ciencias de la Tierra, Escuela Universitaria Politécnica, Universidade da Coruña, 15471 Ferrol, Spain

**Keywords:** rutile and anatase nanotube, sol–gel, hydrothermal, DFT, molecular dynamics, synthesis, co-precipitation, nanobatteries, metal oxide, metallic dopants

## Abstract

In the last decade, TiO_2_ nanotubes have attracted the attention of the scientific community and industry due to their exceptional photocatalytic properties, opening a wide range of additional applications in the fields of renewable energy, sensors, supercapacitors, and the pharmaceutical industry. However, their use is limited because their band gap is tied to the visible light spectrum. Therefore, it is essential to dope them with metals to extend their physicochemical advantages. In this review, we provide a brief overview of the preparation of metal-doped TiO_2_ nanotubes. We address hydrothermal and alteration methods that have been used to study the effects of different metal dopants on the structural, morphological, and optoelectrical properties of anatase and rutile nanotubes. The progress of DFT studies on the metal doping of TiO_2_ nanoparticles is discussed. In addition, the traditional models and their confirmation of the results of the experiment with TiO_2_ nanotubes are reviewed, as well as the use of TNT in various applications and the future prospects for its development in other fields. We focus on the comprehensive analysis and practical significance of the development of TiO_2_ hybrid materials and the need for a better understanding of the structural–chemical properties of anatase TiO_2_ nanotubes with metal doping for ion storage devices such as batteries.

## 1. Introduction

Titanium dioxide is a highly valued substance, used for its many beneficial properties in a variety of industrial products, including electronics, photovoltaics, paints, and food-grade materials. The creation of nanotubes in TiO_2_ phases, as well as their interactions with other metal dopants, biomolecule–polymer interactions, and interfacial processes, are leading to promising results in the fields of renewable energy and medicine. Nevertheless, much remains to be discovered about the properties of TiO_2_ nanotubes, and considering that there are numerous TiO_2_ phases, including rutile, anatase, brookite, hollandite, and metastable monoclinic phases. By stacking TiO_6_ octahedra with warped edges, their crystal forms are created [1]. In addition, the anatase phase is one of the most effective photocatalysts due to its high photocatalytic activity, chemical stability, and negative carrier potential. The photocatalytic activity is affected by physicochemical parameters such as the grain size, crystallinity, surface area to volume ratio, surface texture, and geometry [1,2]. The fundamental drawback of TiO_2_ for photocatalytic applications is that when exposed to visible light, only a very small fraction of the incident photons are utilized [2]. Rutile TiO_2_’s large band gap accounts for this. Recent research on doped rutile TiO_2_ has shown that it can be used in a wide variety of significant applications, such as photonic crystals, advanced ceramics, photoelectrochemical cells, and rutile-type TiO_2_ crystals, which are regarded as promising photocatalysts, particularly in enhancing the photocatalytic activity with the appropriate dopants [3]. Many metal oxide semiconductors (Fe_2_O_3_, TiO_2_, MgO, and WO_3_) are frequently utilized in photocatalytic dye degradation due to their superior chemical stability, low toxicity, and high band gap. Among them, low sensitivity, high oxidizing power, no toxicity, good physicochemical stability, a large band gap, effective antibacterial potential, and UV-induced photocatalytic activity are merely a few of the many qualities that TiO_2_ possesses [4]. Reactive oxygen species (ROS) are produced by metal oxide nanoparticles (MONPs). The band structure of pure MONPs differs from that of MONPs with dopants or defects. Dopant/defect engineering is a successful means of changing the band structures of MONPs and altering the generation of ROS. Some MONPs with dopants or defects also have applications in sensing, catalysis, and energy [5]. Surface-enhanced Raman scattering activity (SERS) may be effectively and steadily produced by non-metallic doping in semiconducting metal oxides, and it also offers a fresh means to investigate the connection between charge transfer (CT) in catalysis and semiconductor SERS performance [6]. The combined characteristics of the codopants in titanium dioxide nanotube (TiO_2_ NT) membranes have been detailed, and these membranes are different from their monodoped counterparts [7]. TiO_2_ NTs have a greater Brunauer–Emmett–Teller surface area (BET) compared to their powder form, which is helpful for one-electron oxidation during photocatalytic processes. The efficiency of photocatalytic processes is, however, constrained by the comparatively large band gap of TiO_2_ (3.2 eV) as a result of the high rate of photogenerated electron and hole recombination [8] (Figure 1). To solve this problem, when the metal ions Ag^+^, Al^3+^, Cu^2+^, Fe^3+^, Mn^2+^, Ni^2+^, V^5+^, and Zn^2+^ were doped into TiO_2_ nanotubes, except for Ag^+^, the creation of crystal defects by ion doping resulted in higher photocatalytic activity of the catalysts, decreased SBET, a smaller band gap (Eg) for TiO_2_, and a reduction in the rate of electron and hole recombination [9]. The doped ions can also operate as flat sites for the trapping of electrons and holes before reacting with H_2_O_2_, OH, and O_2_ to form -OH and O_2_ radicals. The investigations also showed that Mn^2+^ or Ni^2+^ doping had the opposite effect on the photocatalytic activity of the RB removal catalysts, which was enhanced when Ag^+^, Al^3+^, and Zn^2+^ were added to the TiO_2_ NTs. The elimination effectiveness of RB was 98.7% within 50 min in the presence of TiO_2_ NTs doped with Zn^2+^ and calcined at 550 °C [7,9]. TiO_2_ NTs are a form of widely used one-dimensional nanostructure, in addition to 1D (nanorods and nanowires), 2D (nanoribbon, nanosheet, and nanobelt), 3D (branching nanostructures and meso/nanoporous), and the crystal-facet-tailored TiO_2_ nanostructures. Due to their hollow form, NTs have 400 m^2^ more surface area per g than nanowires and higher porosity. According to reports, anatase TiO_2_ nanobelts had specific surface areas and total pore volumes of 119 m^2^g^−1^ and 0.666 cm^3^g^−1^, respectively [10]. Therefore, using NTs as the gas sensor material rather than nanowires is more practical for gas sensors. Within this context, doped one-dimensional TiO_2_ has received a great deal of interest because it is used in catalysis, optoelectronics, biochemistry, magnetic materials, photocatalysis, and electronics, as well as its mechanical properties and potential uses in other fields [1]. Their higher aspect ratios, active surface areas, and larger and more specialized surface areas, and their lower potentials as well as resistances, innovative electrochemical and optical properties, and increased catalytic activity, would make them viable choices. TiO_2_, which is based on transition metals, has been discovered to be a particularly excellent guest object for doping semiconductor materials [1,11]. The recurring crystal phases and coordinated cationic and anionic moieties in the TNT materials provide the materials with a stable, regulated shape. The TNT materials have potential for a variety of chemical, physicochemical, and chemical sensing applications because of their outstanding and superior electrocatalytic characteristics [12,13,14]. Considering the above diversity of existing TNTs, the chemical synthesis of metal oxide NTs has evolved significantly in the last century, and the synthesis of TNT metals/non-metallic dopants has experienced tremendous growth in recent decades since the advent of nanotechnology; from both scientific and technological perspectives, there is an urgent need to review the spectrum of molecular doping in terms of morphology, size, characterization, encapsulation, and nanocomposite behavior history. Therefore, the present review addresses various existing synthesis methods of TiO_2_ NTs in the presence of metal dopants. We have observed the dimension and size of various TiO_2_ NTs and the related physicochemical observations on the deposition of transition metals and their associated ions, such as Fe, Mo, Nb, Ru, Au, Ag, Pt, Cu, Co, Ni, V, Cr, and Mn. We have presented a DFT computational analysis for the doping of TiO_2_ nanoparticles, followed by the experimental synthesis methods of TiO_2_ NTs with metal dopants, the recent trends in the computational costs for the calculation of NT by classical and quantum molecular dynamic potentials, and, in the last section, we detail the future prospects of the synthesis methods for the metal doping of various TiO_2_ nanostructures, such as nanobelts and nanohelices of NT, which are important for renewable energy and other potential biological applications. Although TiO_2_ nanotubes present very valuable mechanical and optical properties, they have the drawback that the forbidden band is relatively large, so metal doping is an effective method to reduce this gap. In this regard, this review analyzes and evaluates the main findings of experimental methods for the metal doping of TiO_2_ nanotubes; among the most prominent methods are the hydrothermal method, sol–gel method, co-precipitation method, and electrochemical anodization method. Similarly, this study discusses the importance of computational simulation as a very promising alternative method, because although DFT has been used as the only method for analysis, classical molecular dynamics is still an unexplored method for the study of doping in metal oxide nanostructures such as TiO_2_ nanotubes.

## 2. Main Experimental Methods for the Synthesis of TiO_2_ Nanotubes

### 2.1. Hydrothermal Method

Hydrothermal synthesis is primarily used to create metal oxide NTs [10]. By using it to create tiny-diameter TiO_2_ NTs, Kasuga et al. [15] developed the hydrothermal approach. Currently, this method is widely used since it is a simple and versatile method; it requires a solvent (H_2_O or another) at a moderate temperature and high pressure. As observed in Figure 1a, the process begins with a precursor solution that can be aqueous or non-aqueous. The suspension is stirred vigorously for 2 h at 50 °C. Then, the suspension is subjected to 130 °C for 22 h in an autoclave. The product obtained must be centrifuged, filtered, and the solid sample must be subjected to an annealing wash at 400 °C, with which the TiO_2_ nanotubes will be obtained.

### 2.2. Self-Assembled Electrochemical Anodizing Method

Self-assembled electrochemical anodization (SOA) allows for the production of nanostructured arrays (such as aligned holes, nanochannels, and NTs) that are vertically oriented, size-controlled, and back-contacted (i.e., bonded to a metallic substrate) [16]. The metal of interest (M) can be used as the anode in a straightforward anode/cathode configuration to achieve anodization. Under the proper voltage, where the supply of oxygen ions is normally H_2_O in the electrolyte, it oxidizes to form a metal oxide as the basis for the oxide growth process; see Figure 1b. Template-assisted growth can produce TiO_2_ NTs with a wide range of diameters and good homogeneity by simply changing the size and shape of the templates. For a procedure with a template, see [17]. Contrary to the electrochemical anodization method, it is compatible with a range of substrates, such as silicon and glass. The most common technique for creating NTs using positive template-assisted growth involves coating the templates’ outer surfaces with TiO_2_ and selectively etching them with wet chemicals. The first electrochemical deposition of TiO_2_ NTs with template-assisted growth was described by Hoyer et al. [18] employing an ordered alumina template with a positive polymer form that was acetone-dissolvable.

### 2.3. Sol–Gel Method

In the sol–gel scheme, a precursor is dissolved in a solvent to produce a suspended emulsion that forms a gel after constant stirring [19]. Once the congruent suspension is attained, composite templates of anodized alumina or membranes made of polymers are soaked in the liquid medium. Further, the templates’ pores are filled with suspended atoms and adopt their patterns; see Figure 1c. The required techniques of drying, calcining, and removing the gels produce the metal oxide NTs [20]. The template on which the particles are deposited, the content of the colloidal solution, the temperature, and the period of deposition all affect the particle size and the dimensions of the nanostructures [21,22,23,24,25]. The usability of the sol–gel nanostructures is constrained by the fact that they are frequently produced in bundled form [26]. The morphology of the final material is also impacted by the procedure needed to split the nanostructure derived from the template substance [26]. The studies on the synthesis of TNT by different methods are very limited, as noted by Shalini et al. [8], who discussed nearly thirty-five different methods for the preparation of TiO_2_ NTs, all of which dealt with the anatase phase.

## 3. Experiments on Doping of Metals on TiO_2_ NTs

TiO_2_ NT has improved properties for photocatalytic applications compared to colloidal and nanoparticulate forms. TiO_2_ NTs contain a higher concentration of OH. The strong photocatalytic activity of TiO_2_ NTs was greatly attributed to both the increased capacity to absorb UV light and the large specific surface area of the NTs. Anatase TiO_2_ NT has a band gap energy of 3.25 eV, which is somewhat higher than both rutile TiO_2_ (3.2 eV) and anatase TiO_2_ (3.2 eV) (3.0 eV). Rutile TiO_2_ modification has a direct energy gap of (3.0 eV) (−0.1 eV) [27], brookite (−3.4 eV) (−0.1 eV), and anatase −4.2 eV. The indirect gap caused by anatase alteration is (3.2 eV) (−0.1 eV). In the context of chemical doping of TiO_2_ NTs, the benefits of TiO_2_ are its lack of toxicity, abundance of resources, and chemical stability. As a result, TiO_2_ has received a lot of interest for a variety of applications, including lithium-ion batteries and biomedical ones [28,29,30,31,32].

TiO_2_ can be used; however, its applications are constrained by its broad band gap and high rate of electron–hole recombination. The photo-response of TiO_2_ must be expanded to the visible light area [33,34,35]; see Figure 2. In order to do this, a number of effective alterations have been made to improve the activity, including doping with metal ions—one of the most popular techniques—and decoration with metals, metal ions, and non-metals. Transition metals (TM) possess several valences and a d-electron structure that is not full in order to accept extra electrons, bring impurity levels into the TiO_2_ band gap, and operate as a shallow trap for photo-generated electrons or holes in order to restrict the recombination of electron–hole pairs [36,37,38,39,40]. Physical methods such as laser ablation, magnetron or thermal sputtering, ion implantation, and so on, as well as chemical methods such as solvothermal, chemical, and electrodeposition (Figure 3a), sol–gel, hydrothermal (Figure 3b), direct oxidation, and so on, are available for the synthesis of doped TiO_2_. Numerous benefits come with the hydrothermal approach, including affordability, ease of use, adjustable nucleation, a suitable reaction time and temperature, and high productivity. The hydrothermal method is particularly attractive because it allows the production of uniform TiO_2_ NTs with an outer diameter of around 10 nm [41,42,43]. Similar to NTs, TNTs have sparked a lot of interest in the biological fields due to their simplicity and the inexpensive cost of manufacture by electro-chemical anodization.

By altering the applied potential, pH, the amount of F ions in the electrolytes, and other variables, it is possible to accurately control their lengths, wall thicknesses, and diameters [44,45,46]. In the context of NT, Sn-doped TiO_2_ (B) NTs had NT capacities of 241.6 mAh g^−1^ after 100 discharge/charge cycles at 0.1 °C, and 115.9 mAh g^−1^ after 10 cycles at 2 °C, according to Li et al. [47]. In order to reduce the Li^+^ transfer distance, Sn^2+^ doping appears to improve the electrical conductivity and contact between the electrode and the electrolyte. According to their research, Sn-doped TiO_2_ (B) NTs may have potential as a negative electrode component for creating reliable lithium-ion batteries. Although the results are promising, using a template-based liquid-phase deposition technique, Tu et al. [48] created Sn-TiO_2_ NTs. They discovered that the NTs had a maximum methylene blue degradation rate of 88% in 6 h and 95% in 8 h when exposed to UV light. The light absorption only reaches the near-UV range due to the quick recombination of photo-generated charge carriers in Sn-TiO_2_ and Sn doping in their research, which nevertheless has an impact on the photocatalytic efficiency. Li et al. [48] introduced an oxygen gap in TiO_2_ as a solution to this issue and a practical method to enhance the functionality of Sn. Although the results are promising, using a template-based liquid-phase deposition technique, Tu et al. [48] created Sn-TiO_2_ NTs. Because of the generation of an intergap containing electronic states as a result of the hybridization of the O 2p orbital with the Ti 3d orbital, the oxygen-coated TiO_2_’s band gap would be reduced as a result of charge transfer from the O 2p orbital to the Ti 3d orbital (Vo-TiO_2_). Figure 4 displays the photocatalytic performance and mineralization rates. Vo-TiO_2_ demonstrated more photocatalytic activity in comparison to TiO_2_, Sn-TiO_2_, and Vo-TiO_2_ due to more effective photoinduced charge separation and the use of visible light by introducing the oxygen gap and doping Sn into the TiO_2_ NTs.

In the presence of an electric current, NTs serve as catalysts. As a result, they can provide molecules that come into contact with the reaction site’s electrons. Anatase has more warped angles between the Ti-O bonds than a 90-degree angle. Due to these configurational variations, anatase has a wider band gap (3.2 eV versus 3 eV) and less thermal stability than rutile [49,50]. However, anatase often has a greater specific surface area than rutile, which is very interesting for (photo)catalytic applications [29]. In fact, anatase and anatase–rutile combinations, such as P25, actually have stronger photocatalytic activity than rutile [46,51,52]. TiO_2_ NTs doped with Zn^2+^ had the highest catalytic activity, according to studies on doping with Zn metals, as a result of the weight fractions of the anatase phase, average crystallite sizes, SBET, energy band gap of the catalyst, and doped ions. These activities decreased as a result of doping with Mn^2+^ or Ni^2+^. According to Figure 5 [53], the TiO_2_ NTs exhibited a hollow, open structure with an average diameter of around 10 nm. The wall thickness of the tubes was also roughly 1 nm.

Despite the fact that there is no known treatment for diabetes, diabetics should have their blood glucose levels closely monitored. Due to their versatility in terms of structural design and composition, ease of separation and storage, high stability, simplicity of fabrication, and tunable catalytic activity, these nanomaterials hold great promise for the development of colorimetric glucose biosensors. This activity is based on the peroxidase-like activity of these nanomaterials when used with glucose oxidase. The TiO_2_ NT and CeO_2_ combination provided by Zhao et al. [55] exhibits the largest concentration of Ce^3+^ and the best peroxidase-like activity when compared to nanowires (NW) and nanorods (NR). The reaction offers a simple approach for the colorimetric detection of glucose and H_2_O_2_ with detection thresholds of 3.2 and 6.1 M, respectively. This is due to the high activity of the CeO_2_/anatase phase of TiO_2_ NT. The details of the three nanostructures are shown in Figure 6 with different doping concentrations of CeO_2_ [55].

Food safety and quality have shown a variety of uses for surface-enhanced Raman spectroscopy (SERS), including the analysis and detection of chemical and microbiological threats [57,58,59]. Recent advancements in SERS techniques for food safety and quality applications have concentrated on the development of enhanced SERS substrates and methods to (1) increase sensitivity and selectivity while reducing matrix interference, and (2) enable nondestructive sampling and in situ detection [57,58,59,60,61]. The activity and reproducibility of the plasmonic material used as the SERS substrate have a strong influence on the quality of the measured SERS spectra. Recently, according to Ambroziak et al. [62], the SERS activity of a layer of cubic silver nanoparticles on a tubular TiO_2_ substrate was found to be eight times that of a typical electrochemically nanostructured silver electrode surface. The deactivation of the active surface was also slowed down by silver nanoparticles on TiO_2_ NT substrates. The active surface had a milder deactivation process, which was also seen for silver nanoparticles placed on a TiO_2_ NT substrate. The geometric dimensions and SEM details of bare TiO_2_ NT and also cubic Ag nanoparticles/nanocubes (AgCNPs) @TiO_2_ NT are shown in Figure 7a,b.

To develop titanium-based implant materials, potentiodynamic polarization tests by Yu et al. [64] show that TNs modified with Au nanoparticles (AuNPs) are superior to unaltered TNs in terms of corrosion resistance. AuNPs are deposited on TiO_2_ NT arrays (TN) by electrochemical deposition to improve the surface properties. The corrosion resistance increases with the amount of charged AuNPs. The adhesion and proliferation ability of osteoblast cells on the AuNP-modified surface of TNT is greater than on the unmodified surface of TNT; this type of AuNP-TNT array can produce titanium-based implant materials to improve bioactivity. Similarly, the electrochemical stability of TiO_2_ NT deposited with Ag and Au nanoparticles was presented by Arkusz et al. [65], proving that it was possible to deposit Au nanoparticles with a size range of 20.3 ± 3 nm on the top, inner, and outer surfaces of TiO_2_ NTs and that doing so did not cause structural damage to the TNT. According to TNT re-crystallization testing, the annealed anatase TNT was stable in aqueous environments and did not experience any significant structural changes, as shown in Figure 8A. According to Figure 8B, TNT has lower conductivity because its EIS displays a larger semicircle than those of the other modified electrodes. The excellent conductivity of AgNPs results in a total increase in conductivity when TNT is modified with them. This is mostly because of the quick electron transfer. Another indication that TNT-infused Au nanoparticles are more conductive than TNT alone is the charge transfer resistance of AuNPs/TNT. Both AgNPs and AuNPs reduce the electrical conductivity without acting as ion traps. The difference between TNT modified with AgNPs and AuNPs in charge transfer resistance (Rct) should be noted. The increase in Rct could have been caused by the steric restrictions that AgNPs and AuNPs on the electrode had induced. Gold nanoparticles are smaller than silver nanoparticles; therefore, the electrostatic barrier causes repulsive interactions between AuNPs and TNT, increasing Rct [65].

According to a study by Kim et al. [66], Cu-coated TiO_2_ NTs are crucial for improving the performance of Li-ion batteries’ electrochemical system. The differential capacitance (dQ/dV versus V) for the first cycle of TiO_2_ NTs in the anatase phase and Cu-coated TiO_2_ NTs at a current density of 50 mA g^−1^ is shown in Figure 9a for TiO_2_ NTs in the anatase phase and Cu-coated TiO_2_ NTs, respectively, via their cell potential versus differential capacitance curves. The pristine and Cu-coated TiO_2_ NTs have a total discharge capacity of 253.9 and 259.4 mA h g^−1^, respectively. The faster response with Li-ions was related to the improved charge transfer behavior, such that the Cu-coated TiO_2_ nanotubes’ irreversible capacity loss was lower than that of the uncoated TiO_2_ NTs. These results support the copper coating of TiO_2_ NTs as an electrochemical performance enhancer. As shown in Figure 9, electrochemical studies revealed that Cu-coated TiO_2_ NTs exhibit better charge transfer behavior, better performance at high cycle rates, and higher discharge capacities than the original TiO_2_ NTs. Additionally, the results of the uniform Cu coating can be seen in Figure 10 as SEM and TEM images. It is observed that anatase NTs have a wall thicknesses of approximately 15 nm and interior diameters of approximately 100 nm. Each TiO_2_ NT in the array in Figure 10a is around 18 μm long and is equally spaced over the surface. The NTs have an inner diameter that varies between 40 and 50 nm, an inner wall thickness of around 30 nm, and an outside wall thickness of roughly 10 nm. Cu is consistently deposited on the surface of TiO_2_ NTs as shown in Figure 10b, in contrast to the original TiO_2_ NTs in Figure 10a. The inner diameter of the Cu-coated TiO_2_ NTs was smaller than that of the parent TiO_2_ NTs as a result of the uniformly coated Cu on the surface of the TiO_2_ NTs [66].

To achieve the efficient completion of the photoelectrochemical water-splitting process, it is possible to insert appropriately tailored nanostructures into the photoelectrode to improve the light–matter interactions for effective charge production, charge transport, and the activation of surface chemical processes [64,67,68,69,70]. In plasmon-enhanced photoelectrochemical water splitting, the high electrochemically active surface area, optical absorption capability, and charge transfer rate play a key role in improving the photoelectrochemical activity of materials, as seen in the work by Cai et al. [71]. In particular, for ZnO-coated TiO_2_ NT, as shown in the SEM image of Figure 9a,b, when light with 420 nm wavelength cutoffs is irradiated, the result is as depicted in Figure 11a,b. In comparison to the naked TiO_2_ NTs, the ZnO-covered TiO_2_ NTs have improved photoelectrochemical activities. In particular, for the 10-cycle ZnO-covered TiO_2_ NTs, a considerable rise in the photoelectrochemical activity is seen. With the 10-cycle ZnO/TiO_2_ NTs, the PC density rises from 1.4 μA/cm^2^ for the naked TiO_2_ NTs to 2.2 μA/cm^2^. In contrast to the bare TiO_2_ NTs, the TiO_2_ NTs coated by a 10-cycle ZnO deposit or by a 2.1 nm ZnO coating exhibit an increase in photoelectrochemical activity of over 60%. The 10-cycle ZnO/TiO_2_ NT photoelectrode still displays quick and outstanding transient photocurrent responses in the PC density (Figure 11c) [71] after around 1.5 h of intermittent visible illumination with a cutoff wavelength of 420 nm.

Due to the physicochemical importance of metal-doped TiO_2_ NT, there are a number of variations and challenges associated with the various metal-doped TNTs. The following table (Table 1) explains the recent trends in the three main synthesis methods, size, phase, and dimension, as well as their characterization and physicochemical parameters, for different metal dopants of TNTs.

## 4. Computational Studies of TiO_2_ Nanotubes

### 4.1. DFT for Materials

Among the conventional computational models, the robustness and generality of first-principles electronic structure methods come at enormous computational expense. The first-principle electronic structure theories or ab initio approaches also refer to these techniques; they rely on the Schrödinger equation being solved, which, from the ground up, defines how electrons and nuclei move in a chemical system [105,106,107,108,109]. These methods calculate the molecular energy and associated properties of a given molecular system based on fundamental physical concepts, such as the electrical and nuclear composition of atoms and molecules. First-principles calculations provide precise evaluations and are always aimed at improving the calculation of the atomic- and electronic-level properties of materials. They are based on quantum physics, electrodynamics, statistical thermodynamics, and classical mechanics. First-principles calculations can now efficiently and accurately solve a wide range of material-related issues—for instance, those relating to the design of materials for electrical power production, automotive applications, energy storage, microelectronics, and the chemical industry generally [107,110,111,112,113]. Even though first-principles computations are time-consuming, the approximations can be improved in various situations, such as transition metal oxides, rare-earth compounds, van der Waals interactions, etc., by using so-called post density functional methods (DFT), which are computationally more difficult but still helpful [54,56,114,115,116]. By solving these quantum physics puzzles, first-principles calculations are kept devoid of any empirical knowledge, and their predictive power is particularly useful when searching for new materials with as-yet-unidentified properties [117,118,119]. The discovery of the actual thermodynamic ground state is aided by the fact that fundamental thermodynamical properties, such as the heat of formation, can be calculated at least as accurately as the results of experiments, but much more quickly and for a much wider range of potential structures and compositions. DFT electronic structure calculations can easily determine the forces on atoms and stresses, which is necessary for improving the structural parameters and determining the vibrational properties. By incorporating phonons, the applicability of DFT research is substantially increased because the majority of the effects of temperature on the free energy can be taken into consideration. These matrix-diagnostic schemes in quantum and classical mechanics provide microscopic details for the system of interest. One such example is that the properties of a doped semiconductor material calculated at DFT have practical potential for increasing the efficiency of a photoelectrochemical cell [120,121,122,123]. Additionally, a 210,000 × 210,000 matrix must be created and diagonalized in order to perform a DFT calculation and obtain the lowest eigenvalues of very large matrices—for example, for the smallest systems of 1000 basis functions with around 300 electrons [124,125,126,127]. This is much too large. This matrix alone demands around 300 GB of storage space on disk. The helical phototranslation symmetry of these structures at NTs was used to compute NT with large unit cells (432 atoms in the case of the (36,36) NT) [128,129]. Automatic structure generation, one- and two-electron integral computation, generation of the complete Fock matrix by rotating only the irreducible fraction of the Fock matrix, and diagonalization of the Fock matrix are all made possible by this symmetry [128,129,130].

### 4.2. Quantum Computational Verification of Experimental Doping of Metals on TiO_2_ Nanoparticles

A 40-nm-diameter cobalt-doped TiO_2_ nanoparticle was created. Based on DFT, it was determined that cobalt doping changed the band structure of TiO_2_ since the band gap of cobalt-doped TiO_2_ was 72% smaller than that of pure TiO_2_. The structure simulation and mechanism analysis have demonstrated the co-doped TiO_2_ gas sensor’s improved characteristics and excellent gas-sensing capabilities at ambient temperature [131]. An ab initio study of the systems Ti_1−*x*_R_*x*_O_2_ (R = Mn, Fe, Co, Ni, Cu for various concentrations of substitutional impurities) in the rutile structure revealed that magnetic moments are present in Fe, Co, and Mn but not in Cu or Ni [132,133]. These calculations reveal an essential fact: doping reduces the energy required for vacancies to occur, resulting in doped systems having more vacancies than undoped systems. These findings are consistent with the idea that oxygen vacancies are crucial to the development of magnetism in doped TiO_2_ and may help to explain the variety of magnetic moments that have been experimentally discovered in samples grown under various conditions [131]. Aspects of the electron paramagnetic resonance demonstrated that the co-precipitated TiO_2_ with Fe ions in place of the Ti ions had increased photocatalytic activity. This was credited to the Fe ions integrated into the TiO_2_ crystal lattice’s synergistic effects of greater visible light absorption and low recombination of electron holes. DFT calculations using the CASTEP package based on the plane-wave pseudopotential approach confirmed the role of Fe in the electronic structure of TiO_2_ [134]. In the DFT calculations, the hybrid B3LYP functional and the double zeta basis set LanL2DZ were used. In terms of photocatalytic activity for 4-nitrophenol degradation, Fe^3+^-doped TiO_2_ outperformed undoped TiO_2_. DFT calculations show that the introduction of new electronic states within the band gap is what causes Fe^3+^-doped TiO_2_ to become visible-light-active [135]. For a DFT computational examination of the formation energies of Si, Al, Fe, and F dopants of different charge states over different Fermi level energies in anatase and rutile, a local density approximation (LDA) study is a very well-liked and affordable technique [136]. The formation energy for F doping is shown to be the lowest at interstitial regions, whereas the substitutional sites of Ti have the highest stability for the cationic dopants. All dopants significantly stabilize anatase compared to the rutile phase, indicating that the conversion of anatase to rutile is prevented in such systems, with the dopants arranged in the order F > Si > Fe > Al according to the strength of anatase stabilization. The Al and Fe dopants act as shallow acceptors, with charge balance achieved by the formation of mobile charge carriers rather than anion vacancies [133,134,137,138,139]. In Si-TiO_2_, the Si dopant acts as an inhibitor of the phase transformation from anatase to rutile; in Al-TiO_2_, the Al dopants act as weak inhibitors of the phase transformation from anatase to rutile, and the Fe dopants act as inhibitors of the phase transformation in TiO_2_ [133]. TiO_2_ (Li-TiO_2_) have been studied by DFT [140]. Li itself is not magnetic; it creates holes in O-2p pi orbitals leading to magnetic moments in TiO_2_. The magnetic moments are delocalized but inhomogeneously distributed in the lattice, with long-range ferromagnetic interactions between them. The inhomogeneous distribution of the magnetic moments and the specific crystal symmetry lead to a distance dependence as well as a direction dependence of the stability of the ferromagnetism of the Li-TiO_2_ system. Calculations suggest that Li-TiO_2_ is a potential material for spintronics [140,141,142]. DFT calculations were carried out to examine the effects of Al or Cu doping on the electronic and geometrical structure as well as the photocatalytic properties of TiO_2_ to support sol–gel-based Cu and Al doping studies, and the result was as follows. By optical measurements, the Cu and Al impurities were found to reduce the optical band gap values of the films produced. Finally, the doping process played an important role in improving the photocatalytic efficiency of the samples. Compared with Cu-doped TiO_2_, the Al-doped TiO_2_ film showed the highest photocatalytic activity [143]. For Gram-negative *E. coli* and Gram-positive bacteria Aureus Staphylococcus, the antibacterial capacity of copper (Cu)-doped TiO_2_ (Cu-TiO_2_) was examined under visible light irradiation [144]. DFT studies showed that Cu^+^ and Cu^2+^ ions were introduced into the TiO_2_ lattice instead of Ti^4+^ ions, causing oxygen vacancies, and these improved the efficiency of the photocatalytic reaction [144,145,146]. Significantly high inactivation of bacteria (99.9999%) was achieved in 30 min of visible light irradiation by Cu-TiO_2_. First-principles calculations based on DFT [147] were carried out to examine the electrical and optical properties of pure rutile TiO_2_ and TiO_2_ doped with tin (Sn) and zinc (Zn). The SCAPS-1D simulator’s DFT-extracted carrier mobility, band gap, and absorption spectra of TiO_2_ were used to gauge how well solar cells performed in relation to the dopant concentration and TiO_2_ thickness. Particularly when compared to the performance of PSCs with Sn-doped and Zn-doped TiO_2_, the functionality of perovskite solar cells (PSCs) with 3.125 mol% maximum power conversion efficiency (PCE) for Sn-doped TiO_2_ is 17.14, compared to 13.70% for undoped TiO_2_. Sn-doped TiO_2_ has 0.63% better PCE than Zn-doped TiO_2_ at the same doping concentration. The geometric structures and electronic properties of Ni-doped anatase and rutile TiO_2_ have been successfully calculated and simulated using a pseudopotential method for plane waves based on DFT. Under O-rich growth conditions, the band gap for substitutional Ni in Ti-doped anatase TiO_2_ shows a slight decrease of around 0.05 eV in comparison to pure anatase TiO_2_ and contains a number of impurity energy levels that may be the cause of the experimental photocatalysis’ red shift in the absorption edge and activity. The excitation energy of the photons decreases as a result of the impurity energy levels. When compared to pure TiO_2_, a roughly 0.05 eV smaller band gap is present [148]. Trends in the electronic structure and magnetic properties of TiO_2_-based strain gauges with impurities of transition metals (Sc, V, Cr, Mn, Fe, Co, Ni, Cu, and Zn) using DFT were assessed. There is a significant magnetic moment in TiO_2_-Fe and TiO_2_-Mn, with values of roughly 4.21 Bohr Mag/cell and 3.48 Bohr Mag/cell, respectively [149]. However, based on the density of states, TiO_2_-Sc and TiO_2_-Ni show no change in magnetic properties compared to pure TiO_2_ [148,150]. TiO_2_ showed that the more frequent oxygen vacancies and high surface basicity of Ni^2+^-doped TiO_2_ contributed to higher nucleophilic attack activity for the hydrolysis of CWA [151]. To support the aforesaid conclusion, the surface structure and calculated charge distribution were obtained. The XRD spectra of NiO crystals, anatase- TiO_2_, with and without distinctive doping of Ni^2+^ (TiO_2_-Ni), were simulated and the corresponding theoretical curves were considered in comparison with the experimental data. The outcomes display that the experimental curves are in ideal agreement with the curves calculated through the DFT method, indicating that Ni^2+^ doping does now no longer affect the crystal structure. The calculations performed with the Materials Studio program package in the CASTEP module show that metals used for doping, such as Co, Ni, Sb, Zn, Ag, and Mn, decreased the band gap of TiO_2_ and shifted the position of DOS downward [149,152,153,154,155]. These effects facilitated the migration of charge carriers and increased the photoactivity. Doping with cerium leads to greater stability of TiO_2_ even at high temperatures, which offers greater advantages in industrial applications [156,157]. Ce and Zr doping of rutile TiO_2_ was investigated using DFT +U and the HSE06 implementation of the hybrid exchange method DFT [156,157,158,159], revealing favorable dopant incorporation, lower oxygen hole formation energies for Ce doping, and higher oxygen hole formation energies for Zr doping [160,161,162]. Depending on the Ce content and the method employed to manufacture the coupled photocatalyst, the photocatalyst Ce-doped TiO_2_ frequently has a lower band gap than TiO_2_ [157,159,163,164], enabling visible light activation. The Ce content of Ce-doped TiO_2_ determines the photocatalytic activity, which typically reaches a maximum when the Ce content is in the range of 0.025–0.6 mol% [157,159,163,164,165]. The CeO_2_/TiO_2_ interface is crucial for photocatalytic activity, because contact between CeO_2_/TiO_2_ influences activity by encouraging efficient charge separation and opening up holes for chemical reaction. For photocatalytic applications, 10% or less ceria is suggested for TiO_2_-CeO_2_ [164].

## 5. Current Demands to Understand the TiO_2_-NT–Metal Doping through Classical and Quantum Molecular Dynamics (MD) to Support Experimental Synthesis and Characterization

Classical molecular dynamics methods make an important contribution to materials design. In reality, a true physical system is made up of numerous components. As a result, MD simulation continues to be a distinctive tool for investigation at the tiny scale. In particular, we must consider the impact of temperature, annealing time, and time between heating steps on crystallization, phase transition, and structure during the doping process in nano-substrates [131]. In the study by M. Predota et al. [166], which does not refer to NT but represents an important approach to understanding the effect of NT, classical MD simulations were performed. To describe the microstructure of the interface between aqueous solutions and the rutile (R-TiO_2_) surface at the (110) temperature and to ascertain the impact of surface charge and hydroxylation on the adsorption of Rb^+^, Na^+^, Sr^2+^, Zn^2+^, and Ca^2+^ ions at the interface between a metal oxide and water, M. Predota et al. [166] used classical MD simulations. Despite the fact that the study was not exclusively focused on NT, their findings from comparing the observations of ion adsorption sites obtained using MD and X-ray techniques show that the charged hydroxylated surface model is superior to the non-hydroxylated model as a good approximation of the actual rutile–water interface. Although there is overall agreement regarding the adsorption sites for all ions and the precise ion positions for Rb^+^ and Sr^2+^, the height of the Zn^2+^ ions above the Ti-O surface for these two sites is significantly different according to the X-ray and MD data. Additionally, in order to make accurate predictions of both the qualitative and the quantitative nature of interfacial phenomena, it has been claimed that the study of the MD results of ion adsorption and their interpretation is essential. Similarly, in the work of Sadaf Shirazi-Fard et al. [167], a study on the encapsulation and release of doxorubicin from TiO_2_ NTs without doping with the anticancer drug doxorubicin (DOX) was presented. According to experimental studies and MD simulations with all atoms, the diffusion coefficients (Di) of DOX molecules are in the order of 10-10 m^2^/s. DOX molecules have numerous H-bonding interactions with TiO_2_ NT walls and water, both short- and long-range. They calculated the strength of hydrogen bonds using radial distribution functions (RDFs) and combined radial/angular distribution functions (CDFs). In their study of Li-ion interaction with TiO_2_ nanostructures, Kerisit et al. [168] used MD and a core–shell potential model to look at Li-ion diffusion in rutile and anatase TiO_2_. It was investigated how electron polarons affect Li diffusion. Li-ions and electron polarons will form strongly coupled pairs at low Li mole fractions. It is demonstrated that the Li diffusion in rutile is 4–5 orders of magnitude faster than that in anatase. In the research by Yildirim et al. [169], MD simulations using the DLPOLY-MD package and DFT are used to assess the energetics and dynamics of Li-ion transport in anatase and amorphous TiO_2_ (Figure 12). An investigation was performed on how diffusivity is affected by the Li-ion concentration. In comparison to anatase, the diffusion of Li is slower in amorphous TiO_2_. According to reports, in amorphous TiO_2_, the highest Li intercalation ratio happens at concentrations of 50% and higher. DFT and MD simulations have been used to examine the impact of the Li-ion concentration on diffusivity in rutile TiO_2_. At concentrations of roughly 50%, the energy barrier is at its lowest.

This means that NTs, with their biocompatibility and chemical stability, are desirable materials for drug delivery systems. When employing TiO_2_ NT as the anode for Li-ion batteries, larger storage capacities and higher charge/discharge rates can be attained. This was demonstrated by Li-ions on anatase NTs in a (MD) simulation [170], as shown in Figure 13a,b for the atomic configuration of a lithium–TiO_2_ NT system; in their work, the simulation of adsorbed Li-ions reached saturation quickly depending on the temperature. The number of adsorbed ions increases when the temperature reaches 1000 K, but the quantity of ions drops as the temperature rises more. Li-ion adsorption does not significantly alter the volume of the NTs. Tetrahedral or octahedral sites are where the Li-ions are adsorbed. The ions jump to the surrounding sites during the simulation because the thermodynamically unfavorable octahedral sites, in contrast to tetrahedral sites, have no effect on the number of adsorbed ions. When the surface coverage is modest, the Li-ion adsorption on anatase NTs follows a Langmuir adsorption curve, resulting in few interactions between the deposited ions. These simulations demonstrate that anatase NTs are good potential anodes for Li batteries, and their findings support this assertion.

TiO_2_ NTs were predicted by Zhenyu et al. [171] to have better electron transport than nanocrystal films. In the context of time-dependent density functional theory (TDDFT), their observations of the quantum-classical approach to non-adiabatic (MD) (NAMD) reveal that oxygen vacancies, which are frequent in TiO_2_, significantly increase non-radiative energy losses. Localized Ti^3+^ states are produced by oxygen vacancies hundreds of meV below the TiO_2_ conduction band. These states encourage higher electron–phonon couplings, trap excited electrons, decrease the NT band gap, and make relaxation easier. These findings support the development of TiO_2_ NTs’ structure and charge transfer. Recently, Lei et al. [172] used MD simulations to examine the electrostatic characteristics of the interfaces between semiconductor TiO_2_ and plasmonic nanoparticles of Ag or Cu. To explore the optical and photoelectrochemical properties, they were deposited on TiO_2_ NT arrays (TiO_2_ R/T) using a two-step pulse electrodeposition technique. The results ensured that the bimetallic system had a large potential drop from the Helmholtz layer simulated by MD. An ab initio study was utilized by Elham et al. [173] to better understand how ruthenium doping and hydrogen passivation impacted the structural alteration and, subsequently, the overall electronic band structure of an anatase TiO_2_ NT (TNT) and its efficient water-splitting capacity. The electronic structure of Ru-doped a-TiO_2_ (001) and H:a-TiO_2_ (001) PscWf and post-processing programs from the Quantum Espresso package were used to calculate and optimize within DFT. A black Ru-doped TiO_2_-x sample with weak electronic coupling between its valance and conduction edge orbitals, which results in lower electron–hole recombination and rationalizes the experimental findings, is explained theoretically by their ab initio model of Ru-doped hydrogenated TiO_2_ NTs, Ru-doped, and a pristine sample. Sergei et al. [174] used a modified B3LYP hybrid-exchange correlation functional in the context of density functional theory to perform all calculations on perfect, both single- and double-walled (SW vs. DW) TiO_2_ NTs in the anatase and fluorite phases; their findings show that substitutional impurities can significantly change the electronic structures of both TiO_2_ and SrTiOO_3_ NTs. For N and S dopants, the difference between the highest occupied state and the lowest unoccupied state falls to 2.4 and 2.5 eV, respectively, and to 2.5 eV for C and Fe dopants in these flawed NTs. Because some impurity levels are located in the region between the two redox potentials, electron–hole recombination occurs. This makes the placement of the band gap edges less favorable for visible-light-driven photocatalysis. On the other hand, important studies on the mechanism of dynamic surface electron transfer have been carried out using Pt/rutile TiO_2_ of 3 nm size and simulation cells at X, Y, and Z = 6.43 nm and 31.78 nm as a type of effective photocatalyst that allows one to demonstrate significant advantages [175], despite the fact that no classical MD has recently been performed for Pt-doped TiO_2_ NT. The extremely small PT nanoparticle with a diameter of 3 nm that was deposited on the TiO_2_ surface could be examined thanks to MD simulation in combination with a reactive force field (REAXFF). Their MD results confirm the experimentally observed method for the regulated protection of highly active sites (such as PT, AU, and PD) on supports, which is essential for the development of efficient and reliable photocatalysts. The force field of optimal atom potentials for liquid simulations was used to simulate lithium + ion doped with ionic liquid (IL) on the TiO_2_-B (1 0 0) surface, and fixing the TiO_2_ substrate preserved the stable structure [176]. In the same way, simulations were carried out to investigate the mechanisms of wetting and regulation—specifically, how does the variation in the Li^+^ concentration affect the interfacial structure between ILS and TiO_2_? What effect does this have on the contact length and angle of the electrolyte droplets on the TiO_2_-B (1 0 0 0) surface? Doping with Li^+^ was predicted to slow down the dynamic wetting process and alter the final shape of the ionic liquid droplet. For this purpose, optimized condensed phase molecular potentials were used for MD (compass) studies that allowed them to investigate the structures and dynamic properties of a 1 M KOH solution on anatase TiO_2_ (001), (100), and (101) surfaces to understand the molecular details of the effect of KOH on the water splitting of the PEC phenomenon [177]. This study confirms that K+ ions prefer to bind to O2C on the TiO_2_ surface. Furthermore, the layering of oxygen and water atoms on TiO_2_ surfaces significantly regulates the kinetics of potassium ions. The first study of substantial amounts of water behavior on the surfaces of anatase (101) nanotubes with a diameter of 1 nm used traditional MD simulations [178], and the results show that water inside the tube diffuses more slowly than water in contact with the outer surface tint. The anatase phase of TiO_2_ NTs would allow them to create nanoscale ion beams and serve in place of the huge magnets currently used for beam steering, according to studies that have recently focused on the relatively restricted channeling property of TNT [179]. Based on the Lindhard planar channeling potential, (MD) simulations were used to examine the channeling of HE++ mega-electron ions (MEV) in titania nanotubes. The ion channeling phenomenon made it possible for titania nanotubes to have a diameter of 100 nm and 2 m and a length of 1 m and 2 m, as shown in the simulated trajectories of ions projected onto them. To study the gas-sensing properties of TiO_2_, Natalia et al. [180] used all MD atoms to model a 150 × 150 surface area of atomic planes of 100 anatase TiO_2_ and TiO_2_:MoO_3_ compounds to study their interactions with water, hydrogen, methane, and ethanol molecules at temperatures of 300 K and 573 K. The synthesis of anatase-modified TiO_2_ and TiO_2_: MoO_3_ composite materials with variable MoO_3_ content by the sol–gel method helped to verify the results of the simulations of molecular dynamics. On the basis of the synthesized oxide materials, single-electrode thermocatalytic chemical gas sensors were constructed, and experimental tests were carried out on their detection abilities for hydrogen, methane, and ethanol gases. According to IR spectroscopy data, the number of surface OH groups and adsorbed water molecules in the TiO_2_:TiO_2_ composite is significantly higher than in bare TiO_2_, and decreases with increasing MoO_3_ content. This experimental finding is reliable with the results of the (MD) simulation on the adsorption of the water molecule on the TiO_2_ and TiO_2_ surfaces. In recent years, the great differences in the properties of nanotubes and other TiO_2_ nanostructures have captured the attention of many researchers, since geometry is decisive in the interaction with other elements, as well as the combination with other materials [181,182,183]. In the same direction, to explore the formation of non-abundant TiO_2_ crystalline thin films (both rutile and anatase), Houska et al. [184] pulverized samples with a magnetron and performed atomic-level MD simulations with a Buckingham interaction potential, using a simulation technique that is iterative. They estimated the effect of the energy of the arriving atom, the structure of the substrate, and the lateral size (growing crystal) and the temperature of the substrate, and confirmed that the phenomena observed experimentally (in compared to anatase, the growth of rutile at higher energies per atom of the film and a higher temperature of the nucleation support temperature) were consistent with the MD results. The results shed light on the intricate relationships that exist between the process parameters and the structures of the TiO_2_ films that are deposited. Using high-resolution transmission electron microscopy (TEM), in situ ion irradiation TEM, and (MD) simulations, amorphous TiO_2_ nanotubes were compared with their crystalline counterparts, anatase TiO_2_ nanotubes. According to the calculations of MD [185], the internal stresses caused by the densification process during crystallization cause partially crystalline tubes to bend. TiO_2_ nanotube-enhanced biomedical devices have been shown to significantly affect mesenchymal stem cell proliferation, differentiation, and adhesion. Cells react to the nanotubes via increasing adhesion, proliferation, and differentiation. In a biological environment, proteins act as an intermediate layer on the surface of the material to further promote cell adhesion and proliferation, and protein adhesion is the first event to occur when the implant surface makes contact with the biological environment (tissue, body fluids). Smaller-diameter TiO_2_ NTs have a greater surface area for the binding of positively charged proteins than larger-diameter NTs [186]. Histone has the most adsorbed protein in 100-nm-diameter nanotubes (10 nm in length), while 15-nm-diameter nanotubes have higher values.

## 6. Conclusions

This review summarizes the common techniques used to synthesize and modify the properties of TNTs. TNTs have great potential for applications in the energy industry as efficient photocatalysts, for photoelectrochemical water splitting for hydrogen production, and as better photoanodes in photovoltaics of solar cells, while poor solubility factors enrich their applications in various biomedical fields. Nanostructured TNTs with factors such as a tunable length, wall density, and pore size result in a large surface area for high-performance solar energy utilization. Here, we have detailed some important observations on metal dopants for TNTs. In particular, thermogravimetric analysis has shown that the co-addition of Co_3_O_4_-TNTs increases the thermal stability of epoxy resin (EP). This implies that Co_3_O_4_-TNTs can effectively increase the flame retardancy of EP and have a good synergistic flame-retardant effect. According to cone calorimeter measurements, EP/Co_3_O_4_-TNTs exhibited the lowest peak heat release rate and had 35.4% lower overall heat release than pure EP. This implies that Co_3_O_4_-TNTs can effectively increase the flame retardancy of EP and have a good synergistic flame-retardant effect. According to calorimetric experiments, TNT-loaded CeO_2_ can significantly increase the thermal stability and flame resistance of matrix materials. CdSe/TiO_2_ loaded with CeO_2_ may effectively slow the release quality and quantity of heat and encourage the creation of a protected carbon layer, which can achieve a flam-retardant effect. After 30 min of immersion in a CdSe precursor solution, CdSe/TiO_2_ NTs displayed a maximum photocurrent density of 0.0016 A/cm^2^, an improvement of around 10% over the original NTs when exposed to sunlight. The adsorption edge of the Ag-modified sample TNT significantly shifted to the blue. The unmodified TNT and Ag_2_0-TNT samples had Brunauer–Emmett–Teller (BET) surface areas of 392 and 330 m^2^g^−1^, respectively. The main challenge for doped TiO_2_ photocatalysts is that the photocatalytic activity is lower in visible light than in ultraviolet light. As a result, more research into these photocatalysts is required. The development of optimized dopants and doping strategies is a critical area of future research. Self-assembled TiO_2_-based NTs are a particularly promising avenue, as alloy growth and the anodization process enable previously unexplored doping techniques. Doping titanium dioxide with metals/non-metals increases the electrical conductivity due to the formation of local energy states within the band structure and/or the formation of lattice defects (oxygen vacancies, Ti^3+^ species). In addition, controlling the variation in the radii of host and doped ions and supporting oxygen depletion can lead to increased stability and activity of TiO_2_(B) during cycling in SIBs and LIBs. Combined with an extremely high degree of geometry control in reactive TiO_2_ systems, experimental and quantum DFT and classical computational studies are currently in demand, and there is still room for understanding the electrochemical performance and various metal-doped TiO_2_ NTs of anatase-based anodes for LIBs and SIBs using such effective nanostructures of NTs and in optimizing the comparative analysis of other anatase geometry states, such as nanorods, nanofibers, nanoribbons, nanowires, or nanoplates, with similar metal doping to produce hybrids and nanocomposites for use in metal-ion batteries and other related renewable energy applications including machine learning to improve classical (MD) simulations without increasing the computational complexity and providing a methodology that could be used to predict other electronic properties of a larger system of NTs with metal doping that cannot be calculated with classical simulations alone. Due to their high aspect ratio, exceptional flexibility, elasticity, and optical qualities, TiO_2_ nanohelices (NHs) have garnered a lot of attention. These characteristics give rise to promising performance in a wide range of crucial domains, including optics, electronics, and micro/nanodevices. However, creating spatially anisotropic helical structures from stiff TiO_2_ nanowires (TiO_2_ NWs) is still difficult. To assemble individual TiO_2_ Nws into a DNA-like helical structure, a pressure-induced hydrothermal method was developed [187]. Vertical TiO_2_ nanohelix arrays were created by intertwining synthesized TiO_2_ NHs (50 nm in diameter, 5–7 mm in length) with TiO_2_ NHBs (20 nm in diameter) (NHAs). Thus, theoretical calculations also supported the finding that straight TiO_2_ NWs preferentially change into helical conformations with the lowest possible entropy (S) and free energy (F) to continue growing in a small area. The excellent elastic characteristics have a lot of potential for use as buffer materials or in flexible devices. The creation of new hybrid materials will be greatly aided by additional computational and experimental verification of the NT nanohelices of TNTs, as with the doping of NT.

In the same way, another relevant aspect is that the classical (MD) simulations that are currently available and are experimentally challenging include the formation of nanotubes in the aforementioned TiO_2_ phases and their interactions with other metal dopants, as well as the interactions between biomolecules and polymers and interfacial aspects. We believe that in experiments with various doping elements, such as DFT research, similar processes can be replicated using classical MD. Furthermore, a modern technique for simulating Ni-doped MoS_2_ to understand the effect of doping was found to be the reactive force field (ReaxFF), connecting quantum mechanical and experiments [188]. This makes TNT metal doping using ReaxFF a particularly useful tool to understand the dynamics of doping research [189].

## Figures and Tables

**Figure 1 materials-16-03076-f001:**
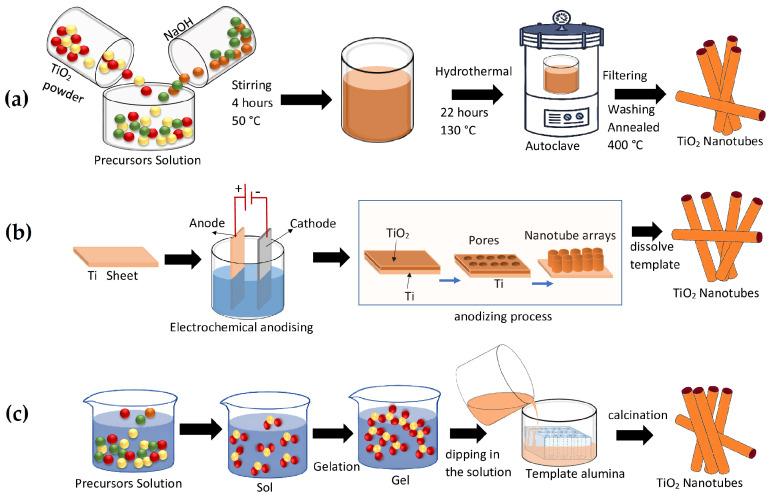
Schematic representation of the main production methods of TiO_2_ nanotubes. (**a**) Hydrothermal method. (**b**) Self-assembled electrochemical anodizing method. (**c**) Sol–gel method.

**Figure 2 materials-16-03076-f002:**
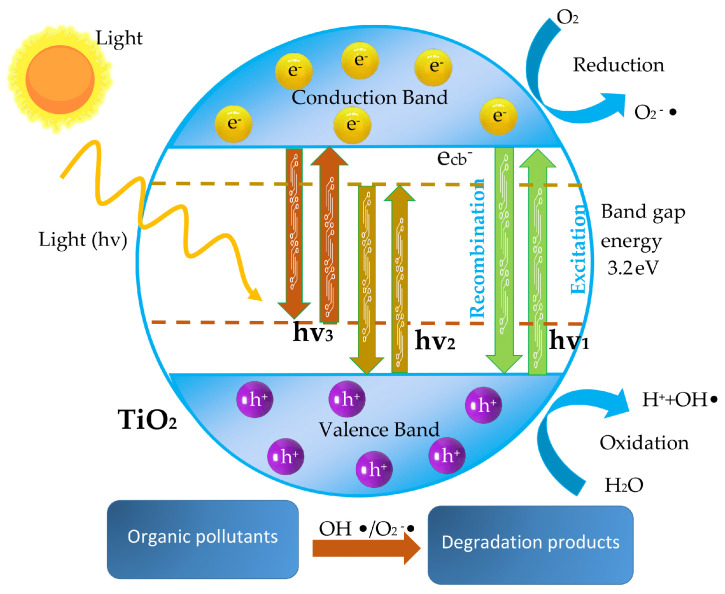
Mechanism of TiO_2_ photocatalysis—hv_1_: pure TiO_2_; hv_2_: metal-doped TiO_2_, and hv_3_: non-metal-doped TiO_2_.

**Figure 3 materials-16-03076-f003:**
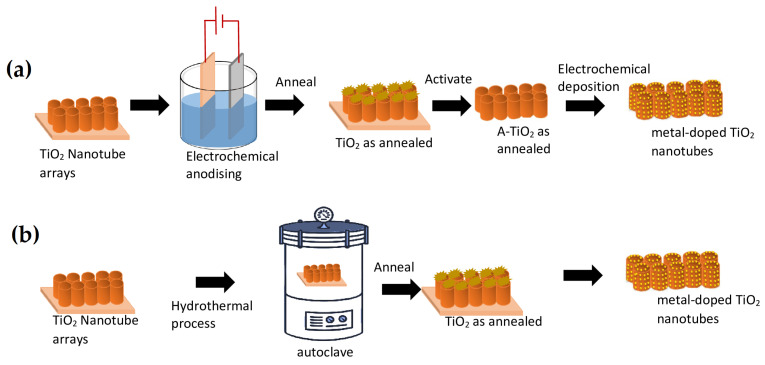
Schematic representation of the most used methods to dope TiO_2_ nanotubes using metals: (**a**) Self-assembled electrochemical anodizing method (**b**) Hydrothermal method.

**Figure 4 materials-16-03076-f004:**
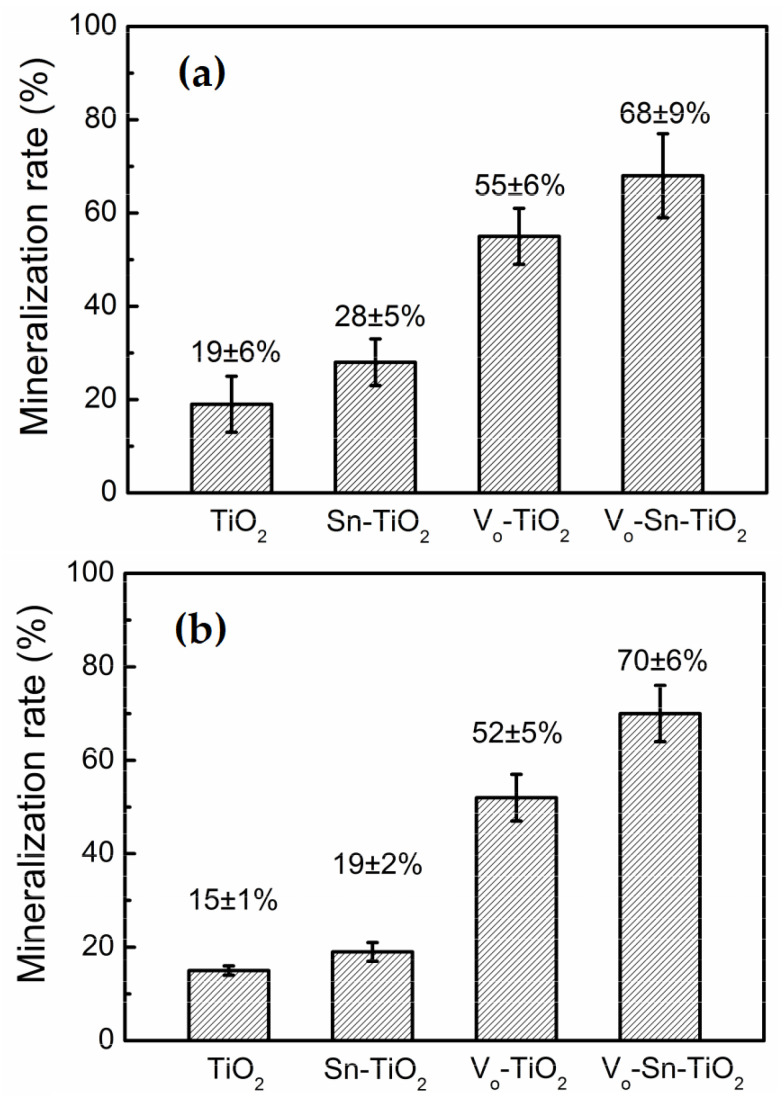
(**a**) Rates of nitrobenzene photocatalytic mineralization by visible-light-exposed TiO_2_, Sn-347 TiO_2_, Vo-TiO_2_, and Vo-Sn-TiO_2_. (**b**) Rates of RhB photocatalytic mineralization when exposed to 348 TiO_2_, Sn-TiO_2_, Vo-TiO_2_, and Vo-Sn-TiO_2_. Ref. [48] is reprinted with permission. 2016 Elsevier B.V. 349 Copyright.

**Figure 5 materials-16-03076-f005:**
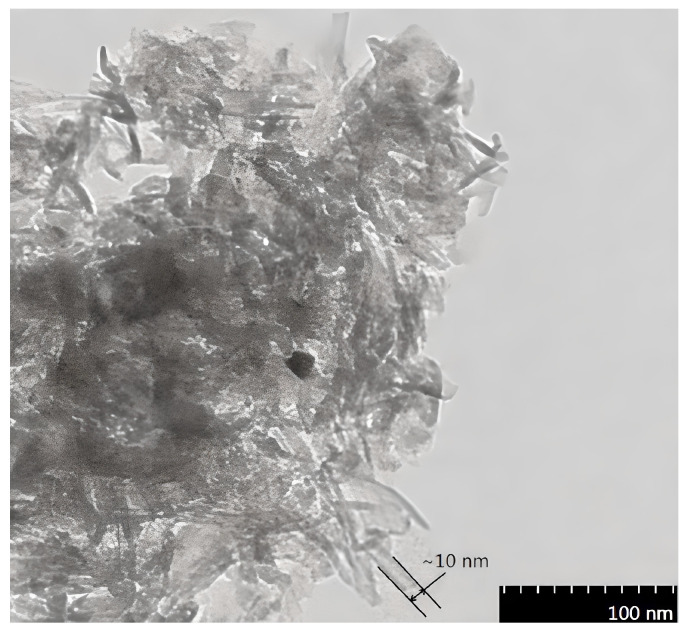
Images of 550 °C calcined Zn^2+^-doped transmission electron micrographs (TEM), reprinted with permission from Ref. [54]. Korean Society of Environmental Engineers, 2015.

**Figure 6 materials-16-03076-f006:**
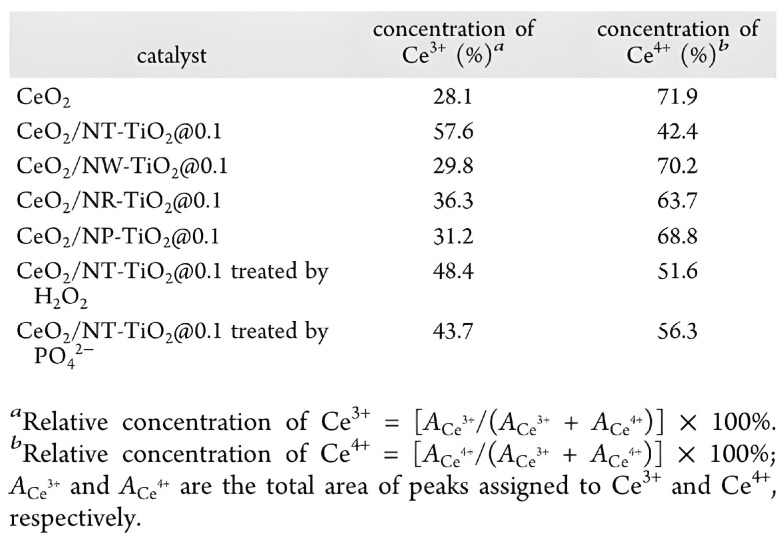
XPS spectra were used to calculate the concentrations of Ce^3+^ and Ce^4+^, which confirmed that the CeO_2_ NT had the best peroxidase-like activity. The presence of TiO_2_ promotes higher Ce^3+^ concentrations in the following order: CeO_2_/NT-TiO_2_@0.1 > CeO_2_/NR-TiO_2_@0.1 > CeO_2_/NP-TiO_2_@0.1 > CeO_2_/NW-TiO_2_@0.1. In comparison to CeO_2_/NW-TiO_2_, CeO_2_/NR-TiO_2_, and CeO_2_/NP-TiO_2_. Reprinted with permission from Reference [56]. American Chemical Society 2015, all rights reserved.

**Figure 7 materials-16-03076-f007:**
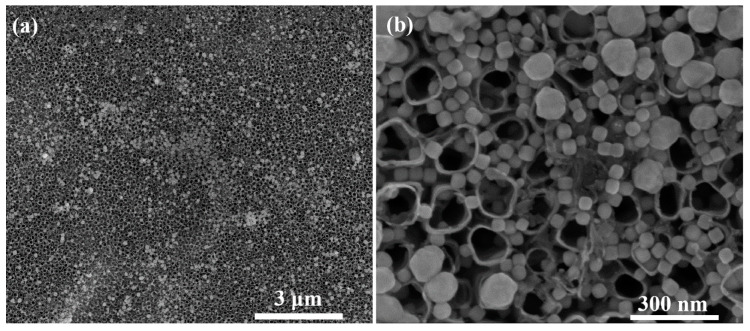
(**a**) SEM at low (**b**) and high magnification: images of cubic Ag nanoparticles/nanocubes with an edge length of around 45 nm deposited by the droplet method on TiO_2_ NT in anatase form with an inner diameter of around 110 nm and a wall thickness of around 20 nm are shown. SERS spectroscopy is one of the most sensitive analytical tools available and is widely used in medicine. Ref. [63] is reprinted with permission. Copyright © 2019, MDPI AG.

**Figure 8 materials-16-03076-f008:**
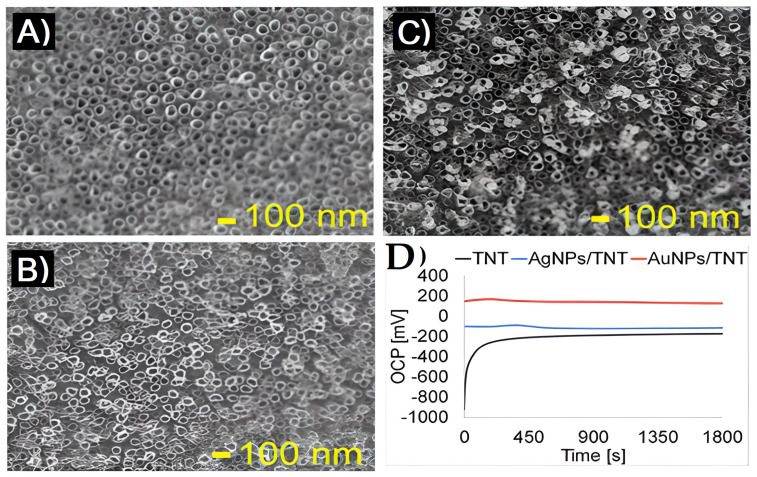
In the anatase phase of TiO_2_ NTs, SEM analysis revealed a uniform, vertically oriented layer of NTs with a diameter of 50 ± 5 nm and a height of 1000 ± 100 nm nm, m, a surface area of 30.27 cm^2^, and a TNT density of 4.26 g/cm^3^ (**A**–**C**). No breaking or delamination of the TNT layer occurred throughout the 2-hour annealing process, which was performed in an argon atmosphere. (**A**) Homogenous layer of vertically ordered, 50 nm diameter TNTs. (**B**) AuNPs of 15–50 nm, (**C**) FE-SEM doped with silver nanoparticles that are spherical and have a diameter between 5 and 40 nm, with 75% of the particles falling between 20 and 40 nm, (**D**) electrochemical impedance data (EIS) for TNT, AgNPs/TNT, and AuNPs/TNT measured in PBS, as well as the Nyquist impedance plots of the analyzed electrodes in the frequency range of 0.1 Hz to 100 kHz. With permission from Ref. [65] 2017 Elsevier B.V. All rights reserved.

**Figure 9 materials-16-03076-f009:**
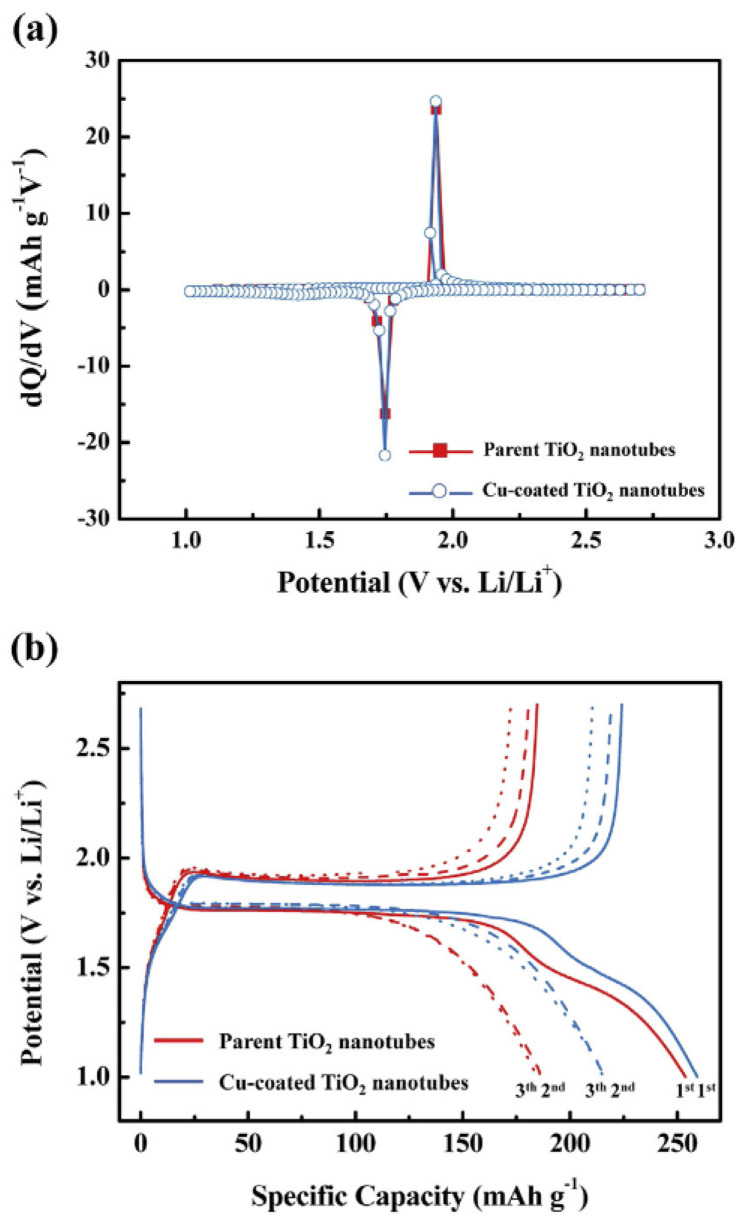
The differential capacity plot (dQ/dV vs. V) corresponding to the first cycle of the parent and Cu-coated TiO_2_ NTs at a current density of 50 mA g^−1^ is shown as (**a**) differential capacity vs. cell potential curves of parent and Cu-coated anatase-phase TiO_2_ NTs at 50 mA g^−1^. (**b**) Discharge–charge curves of the parent and Cu-coated TiO_2_ NTs between 2.7 and 1.0 V at a constant current density of 50 mA g^−1^ show that the Cu-coated TiO_2_ NTs’ Coulombic efficiency was 13.7% higher than that of the parent TiO_2_ NTs. The aforementioned findings point to a more pronounced charge transfer response in the Cu-coated TiO_2_ NTs during the charge–discharge procedure than in the parent TiO_2_ NTs. With permission from Ref. [66]. 2015 Elsevier B.V. All rights reserved.

**Figure 10 materials-16-03076-f010:**
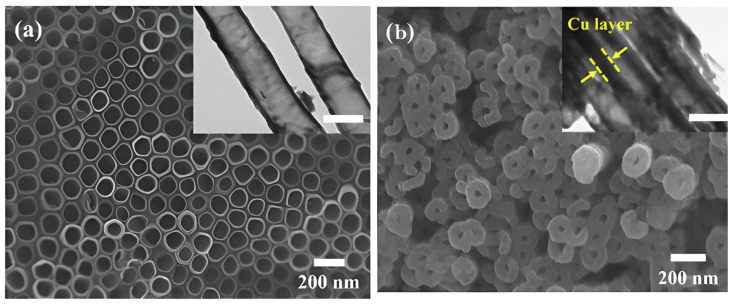
SEM and TEM (inserts) images of parent TiO_2_ NT. According to the SEM and TEM images, the TiO_2_ NT array shown in (**a**) is uniformly distributed on the surface, with the lengths of the TiO_2_ NTs. (**b**) Cu is uniformly deposited on the surface of the TiO_2_ NTs. Reprinted with permission from Ref. [66]. Copyright © 2015 Elsevier B.V.

**Figure 11 materials-16-03076-f011:**
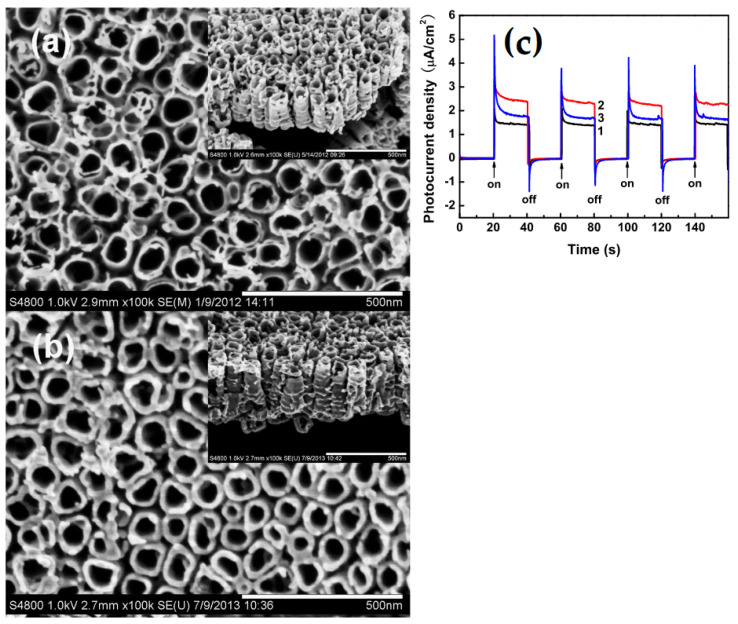
(**a**,**b**) Top view and cross-sectional pictures of bare TiO_2_ NTs, 10-cycle ZnO/TiO_2_ NTs, and 25-cycle ZnO/TiO_2_ NTs with TiO_2_ NTs having average diameters of around 60 nm and wall thicknesses of around 15 nm, respectively. (**b**) When illuminated by light with a 420 nm wavelength, compared to the bare TiO_2_-NTs, the ZnO-coated TiO_2_ NTs exhibit increased photoelectrochemical activity. The 10-fold ZnO-coated TiO_2_ NTs in particular show a notable increase in photoelectrochemical activity. According to PC density (**c**), the 10-fold ZnO/TiO_2_ NT photoelectrode still exhibits rapid and excellent transient photocurrent responses under intermittent illumination after roughly 1.5 h of visible illumination with light of a 420 nm wavelength, as can be seen at PC densities transients of bare TiO_2_ NT (black line), 10-cycle ZnO/TiO_2_ NT (red line) and 25-cycle ZnO/TiO_2_ NT (blue line). Reprinted with permission from Ref. [71]. Copyright © 2023 BioMed Central Ltd., SpringerOpen.

**Figure 12 materials-16-03076-f012:**
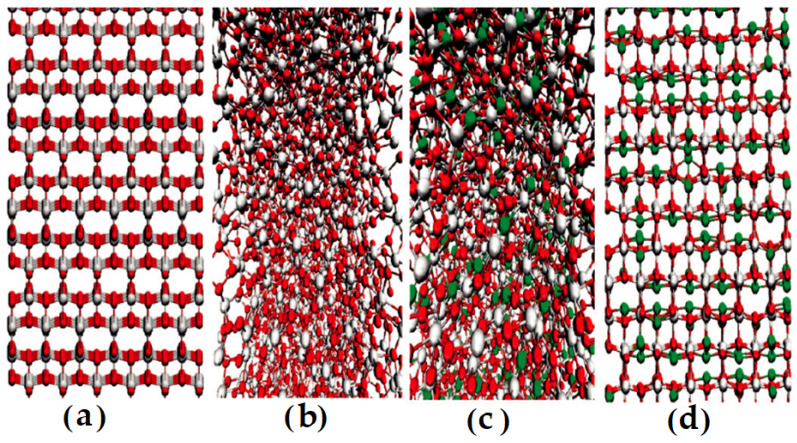
Structures containing 4320 atoms used in the simulations for (**a**) TiO_2_ anatase, (**b**) TiO_2_ 608 amorphous, (**c**) 50% Li-loaded amorphous, and (**d**) 50% Li-loaded anatase. Red, gray, and green spheres represent oxygen, titanium, and lithium, respectively. Reprinted with permission from Ref. [169]. Copyright © 2011, American Chemical Society.

**Figure 13 materials-16-03076-f013:**
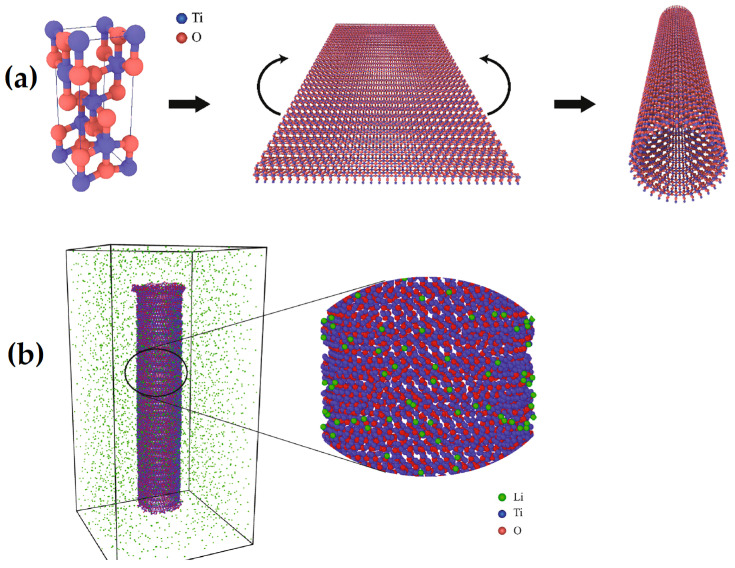
(**a**) Construction of anatase NT (red spheres: Ti, blue spheres); (**b**) simulation box (red spheres: Ti, blue spheres: O, green spheres: Li). Reprinted with permission from Ref. [170]. Copyright©2021 Elsevier Ltd.

**Table 1 materials-16-03076-t001:** Different methods of experimental synthesis, characterization, and their applications in metal doping for TiO_2_ nanotubes.

Pap. No.	Experimental Synthesis	TiO_2_ Anatase/Rutile/Bulk, Size, Interacting Doping System	Observation	References
1	24-h hydrothermal process at 150 °C.	Single wall, the length is up to 140 nm, and 9.3 and 4.02 nm, respectively, are the average values for the outer and inner diameters. Both Ni and Ni/Cr	The transition metals nickel and chromium have complex electron shell structures. When doping was increased, the tubular shape vanished and a sheet-like structure was seen instead. The addition of Ni and/or Cr ions would actively alter the physical characteristics of TiO_2_ by generating impurity energy levels. The estimated energy band gaps of 15% Ni-doped TiO_2_, 6% Ni/4% Cr-doped TiO_2_, and pure TiO_2_ are 2.73 eV, 3.16 eV, and 2.45 eV, respectively. The Ni/Cr-coded TiO_2_ nanoparticles showed a greater degradation efficiency of 96% after 90 min and outstanding stability for five degradation cycles in the photocatalytic investigation. The structural conversion of TNTs into nanosheets is a workable technique that permits the creation of a new category of effective and affordable reusable nano-photocatalysts based on co-doped TiO_2_. The Ni impurity atoms have a significant impact on the Raman modes by altering the vibrations of the TiO_2_ lattice. The large shifts in the Raman spectrum may be caused by the perturbation of the phonon states of the titanium dioxide’s lattice modes and the vibrations of the dopant atoms.	[72]
2	24-h hydrothermal process at 150 °C.	Anatase, length > 100 nm, outer and inner diameter 10 nm and 4 nm, Cu^2+^	Cu^2+^/F-encoded TiO_2_ NTs have a surface area that is greater than both undoped and Cu^2+^-only doped TiO_2_ NTs. Due to the incorporation of copper into the interstitial TiO_2_ lattice sites, the lattice parameters and cell volume of the Cu^2+^-doped TiO_2_ NTs were markedly different from those of the undoped TiO_2_ NTs.	[73]
3	One-step hydrothermal	Anatase, length < 215 μm, the diameter < 120 nm (117, 87, 102 nm), Co^2+^/Co^3+^	Co-black TNTs are highly effective at degrading organic pollutants and activating peroxymonosulfate (PMS). Ti-2p1/2 and Ti-2p3/2 have binding energies of 464.8 and 458.9 eV in bare and Co-TNT, respectively. Hydrothermal treatment of TiO_2_-NTs in CoCl_2_ electrolyte leads to an increase in the actual surface area of the tubular structure.	[74]
4	One-step hydrothermal method	Amorphous, length of 100–200 nm, diameter of 8 ± 2 nm. Si	Si-TNTs have a uniform NT doped with 10% Si, whose photocatalytic methylene blue degradation efficiency triples compared to the undoped TiO_2_ TNTs under ultraviolet light. Si-doped TiNTs are a promising substance for wastewater treatment in the industrial sector. FTIR confirms that the formation of the Si-O-Ti bonds contributes to the stabilization of the lattice and positively affects the photocatalytic activity of TiO_2_.	[75]
5	Ultrasonic-assisted sol–hydrothermal method	Anatase, 20 nm < length < 100 nm. Uniform values of diameter and outer diameter of ≈60 nm and ≈8 nm, respectively. Fe	Higher photocatalytic activity than pure NTs, and *R_Fe_* = 1.75 was found to be the ideal doping concentration. By adding Fe, NTs’ absorption edge was moved into the visible light spectrum, narrowing the band gap. When exposed to visible light, the absorption increased 2-to 4-fold above pure NTs. FeCu^3+^ (upper) ions changed the composition of the phase and modified the catalyst’s surface area, surface area distribution, and photocatalytic activity.	[76]
6	Ultrasonic-assisted sol–hydrothermal method	Anatase, rutile, average diameter 70–90 nm, wall thickness of 10–20 nm. SnO_2_-Sb	SnO_2_ NTs that have Sb doping. With a shelf life of 116 h, the TiO_2_ NTs/SnO_2_-Sb electrode outlasted the Ti/SnO_2_-Sb electrode by a considerable amount (1.6 h). In comparison to electrodes made using the traditional dip-coating technique, those made utilizing hydrothermal synthesis have substantially better and larger SnO_2_ crystals.	[77]
7	Combination of sol–gel process with hydrothermal treatment	Anatase, length > 100 nm, outer diameter of approximately 10 nm Fe	With anatase Fe/TiO_2_ NT, a greater photocatalytic effect was attained compared to Fe-doped NTs. The best photocatalytic activity was found in the 0.5% Fe/ TiO_2_ NTs that were calcined at 300 °C; this activity was superior to that of the pure powder. Methyl orange degradation rate for pure powder and NT after two hours of calcination at 300 °C. 63.5% of the methyl orange decayed in the presence of pure NT after 3 h of irradiation, compared to 69.1% of the methyl orange in the presence of pure powder.	[78]
8	Prior to using the hydrothermal procedure for NT, the sol–gel method was used to create their precursory nanopowders.	Anatase, length of > 100 nm. Inner hole diameter of 5 nm, outer diameter of 10–15 nm. Ce.	TNTs have a 20 nm pore size distribution, with the majority of them falling between 2.0 and 9.0 nm. The creation of the hysteresis loop, which diminishes marginally with increased Ce addition, is adversely affected by the greater pore size dispersion. The 2.5 mol% Ce-doped TNT had the greatest specific surface area (196.01 m^2^/g), lowest zeta potential (0.49 cm^3^/g), largest pore volume (0.49 cm^3^/g), and the best capacity to photocatalyze the dye MB quickly and remarkably effectively even when exposed to sun light. Ce is entirely dissolved in the titanium lattice during the hydrothermal synthesis as Ce^3+^ and Ce^4+^, whose ratio rises with increasing Ce addition.	[79]
9	Two-step hydrothermal method	Anatase, length of > 100 nm, tubular with outer diameters of 8 to 12 nm. Ag	The XRD patterns of the Ag-doped NTs show that the nanocrystalline anatase structure is preserved after doping. After Ag doping of the NTs, the FTIR spectrum hardly changes. In line with the findings of the XRD and FTIR spectra, the Raman spectra for both undoped and Ag-doped NTs confirm the anatase phase. The Ti-O vibrations of anatase TiO_2_ are represented by the Raman bands at 144, 395, 515, and 635 cm^−1^. The characteristic Raman peaks of the NTs become more potent with Ag doping. TiO_2_’s particle size, crystal structure, specific surface area, and morphology can all be altered by metal doping. The increased crystallinity of the Ag-doped NTs may be the reason that their Raman peaks are more intense than those of the undoped NTs. Ag-doped NTs are promising as effective antimicrobial reagents that are free of resistance.	[80]
10	Hydrothermal method	Face-centered cubic (fcc) anatase TiO_2_ (101), Au (111) planes	Platinum island effect in evenly Au-doped NTs aids in increasing isolation efficiency by preventing photogenerated electron and hole recombination. The Au unevenly doped NTs exhibit the “platinum island” effect, or separation of charges at the interface, as a result of the uneven doping. When it comes to the photocatalytic degradation of organic dyes, the non-uniform doping (with Au) of the NTs is superior to the uniform doping. The electrons generated at the interface between pure TiO_2_ and Au-doped TiO_2_ are concentrated around the Au-doped TiO_2_ due to the electrophilicity of Au, leaving a significant amount of holes around the pure TiO_2_. Electrons and holes are generated when UV light excites the non-uniformly Au-doped NTs.	[81]
11	Direct hydrothermal	Anatase and rutile, NTs range in length from tens to hundreds of nanometers, 4 nm < inner diameter < 4.4 nm, 9 nm < outer diameter > 11 nm. Fe	Fe does not form a solid solution with TiO_2_, and these metals are in the interlayer space. The structure of Fe/TiO_2_ NTs is similar to that of Na_2_Ti_2_O_4_-(OH)_2_	[82]
	Direct hydrothermal	Anatase and rutile, Ni	Ni does not form a solid solution, and these metals are found in the interlayer region.	[82]
	Direct hydrothermal	Anatase and rutile, Zn	Zn and TiO_2_ do not combine to create a solid solution, and these metals are located in the interlayer region. Similar to the structure of Na_2_Ti_2_O_4_, Zn/TiO_2_, TiO_2_ NTs(OH)	[82]
	Direct hydrothermal	Anatase and rutile, Cd	In the interlayer structure of Cd/ TiO_2_ titania, a mixture of anatase and rutile phases, Cd and TiO_2_ do not form a solid solution.	[82]
	Direct hydrothermal	Anatase and rutile, Mn	These metals are located in the intermediary layer and Mn does not combine with TiO_2_ to produce a solid solution. Mn/TiO_2_, TiO_2_ NTs have a structure that is similar to Na_2_Ti_2_O_4_-(OH)_2_.	[82]
	Direct hydrothermal	Anatase and rutile, Ni	Ni/TiO_2_ NTs have only four vibrational modes, with peaks at 156.5, 277.8, 400, and 600.8 cm^−1^. The first three modes are consistent with brookite modification of TiO_2_, and the fourth mode is consistent with rutile modification of TiO_2_.	[82]
12	After creating Nd-TiO_2_ powders using the sol–gel method, Nd-TiO_2_ NTs were created using the hydrothermal process.	Anatase and rutile, approximately 10-20 nm in diameter, with a length of 100 to 300 nm. Nd	TEM study reveals that Nd-TiO_2_ NTs have a 100-300 nm length and a diameter of 10-20 nm. Anatase crystallites were found in 0.3% of the Nd-TiO_2_ NTs, with few rutile crystallites. Doping with neodymium increased the visible light absorption of Nd-TiO_2_ NTs, caused a red shift in Nd-TiO_2_ NTs compared to TiO_2_ NTs, and prevented the phase transformation of anatase to rutile. The photocatalytic functions of Nd-TiO_2_ NTs were improved by Nd doping. NTs with 0.3% Nd-TiO_2_ were the most active photocatalytic materials.	[83]
13	Hydrothermal treatment	Anatase, diameter 5 < diameter < 12 nm, 200 nm < length < 500 nm. Fe^3+^ Ni^2+^ Mn^2+^	A superconducting quantum interference magnetometer was used to examine the magnetic characteristics of titanate nanotubes (NTs) doped with Fe^3+^, Ni^2+^, and Mn^2+^ ions (SQUID). The as-prepared NTs changed as the calcination temperature exceeded beyond 350 K; they then changed again when the temperature rose, becoming a mixture of anatase and rutile titanate. The titanate NTs doped with Fe^3+^/Ni^2+^/Mn^2+^ ions behaved para-magnetically.	[84]
14	Hydrothermal treatment	Anatase and rutile, 300 nm in length, Fe on-Pt-TiO_2_ NT	The rutile NTs collapsed, leaving rutile crystals that were around 50 nm in size, whereas the anatase NTs maintained their structural integrity. In comparison to the unaltered material, the hydrogen evolution rate for anatase NTs was six times higher.	[85]
15	A straightforward sol–gel procedure, followed by an alkali–thermal reaction, and finally an NH3 thermal treatment	Anatase, diameter of 6 nm and a length of approximately 160 nm, Ni	Due to its superior rate performance, higher initial Coulomb efficiency (65%), high reversible capacity (303 mA hg^−1^ after 500 cycles at 50 mA g^−1^), and superior long cycle life, the Ni-N/TNT electrode significantly outperforms the TNT and Ni/TNT electrodes in terms of sodium ion transport and storage (8000 cycles). Polarization can be lessened by co-doping with nickel and nitrogen. To improve the electrochemical performance of SIBs, co-doping with Ni and N can successfully tune the sodium ion diffusion rate and the electronic and phase structure of TiO_2_.	[86]
16	One-step solvothermal method	Anatase, diameter is around 0.1–1 μm, length is around 2–10 μm Si	Si in TiO_2_ exhibits more absorption than TiO_2_ NTs, by 5%. Due to doping, Si-TiO_2_ NTs require less energy than other materials to excite electrons from the valance band to the conduction band. TiO_2_ NTs are inferior to Si- TiO_2_ NTs in terms of photocatalytic activity.	[87]
17	Anodizing and for TiO_2_ NT and ion implantation for doping	Anatase, rutile; at 400 °C, anatase had a size of around 30 nm, and as the temperature rose, it gradually grew to 65 nm in rutile, which had a size of 35 nm. At 1000 °C, it grew quickly and flattened out with a size of 70 nm. Anatase NTs with an 80 nm diameter. Cr.	Before thermal treatment, the anodic TiO_2_ NT arrays were amorphous, but they eventually crystallized to anatase at 400 °C and started to lose some of their abundance, although they lasted up to 1000 °C. Rutile started to form in modest amounts at 600 °C and its abundance quickly grew at 800 °C, which is consistent with anatase’s rapid decomposition. Anatase is known to be metastable, and by encouraging defects such as vacancies, the presence of Cr doping expedited the synthesis of anatase and rutile at a lower temperature. The conversion rate of anatase to rutile was significantly reduced by lowering the crystallization temperature of rutile from 600 to 500 °C and of anatase from 400 to 400 °C. After annealing at 1000 °C, the rutile produced in TiO_2_ was extremely stable, while it became unstable in Cr-doped TiO_2_, which broke down to TiO_2_ at 900 °C. The presence of a Cr ion composition in doped TiO_2_ was confirmed by ion beam analysis performed by RBS.	[88]
18	First anodization was carried out under a voltage for 60 V with 30 min and second anodization at 30 V for 30 min (two-step TiO_2_ NTs).	Anatase, there are 68 NTs in the hexagonal base. The lower layer has a diameter of 30 nm and a thickness of 15 nm, while the upper layer is 150 nm in diameter and 25 nm thick. The TiO_2_-NTs are around 1 m in length. The TiO_2_ NTs have a tube diameter of around 70 nm and a thickness of around 15 nm. Fe^3+^	The photocurrent of Fe^3+^/TiO_2_ increased in comparison to the undoped NTs, and the band gap displayed a considerable red shift, demonstrating the higher photocatalytic activity. Fe^3+^/TiO_2_ nanotubes degrade at a rate that is 1.62 times greater than that of two-level nanotubes and 2.17 times more than that of single-level nanotubes. The Fe^3+^/TiO_2_ NTs show much improved photocatalytic activity, a promising catalyst for the breakdown of NB in wastewater, at the optimal doping concentration of 0.1 mol/L.	[89]
19	Anodic oxidation method for NT, and Au was deposited onto the TiO_2_ NTs using the deposition–precipitation method	Anatase Tubular TiO_2_ of 25 nm in diameter Au and Pt	The operating temperature is lowered by doped Au nanoparticles. Compared to Pt-doped TiO_2_, Au-doped TiO_2_ NT offers greater stability and anti-sulfur properties. Au-doped sensors have better repeatability than Pt-doped TiO_2_ sensors. Au-TiO_2_ loses its ability to detect gasses after being exposed to SO2 roughly 20 times, but Pt-TiO_2_ stops working quickly after the second exposure, clearly showing the phenomenon of sulfur poisoning. Overall, a comparison of the Au- and Pt-doped TiO_2_ sensors’ technical recovery curves reveals that the Au-doped sensors have greater repeatability and anti-sulfur capacity. This is because Pt readily binds to sulfur, unlike Au, which does not.	[90]
20	Hydrothermal	Anatase and rutile. Hollow, open-ended, 8-nm-diameter structures on average. Fe and Al	TiO_2_ has more photocatalytic activity as a result of Fe doping. The catalytic activity depends on the concentration, doping ions, and calcination temperature. The maximum catalytic activity was observed in the 550 °C calcined 1.0% co-doped (Al:Fe 0.25:0.75) material. Except for the 5.0% Al- and 5.0% Fe-doped TiO_2_ that showed both the anatase and rutile phases when calcined at 600, the anatase phase was the most common type of TiO_2_ NT. The rutile phase did not transition from the anatase phase in undoped TiO_2_. By increasing the Al and Fe doping concentrations, the development of the rutile phase was dramatically accelerated. When the TiO_2_ NTs were doped with metal ions, the surface defect density of the TiO_2_ NTs rose, which aided in the phase shift. In the ion-doped TiO_2_ NTs, the quantity of oxygen vacancies on the anatase grains’ surfaces increased. Al^3+^ and Fe^3+^ cations were found at substitutional places in the crystal lattice, according to XRD and XPS. SBET dropped as the calcination temperature and ion doping were increased.	[91]
21	Sol–gel method for Sn-doped TiO_2_ nanoparticles, hydrothermal treatment for the powder that was finally annealed at 400 °C for 2 h under flowing argon to obtain Sn-doped TiO_2_ NT	Anatase, diameter of around 10 nm and average length of 150 nm. Sn	The fact that Sn-doped TiO_2_ NTs appear to have a bamboo joint structure is supported by Sn-doped TiO_2_ NTs with varying Sn content (3, 5, and 7 at.%). After tin doping, the NTs have a better distribution. Doping greatly increases the electrical conductivity when the tin concentration is less than 5 atomic% by increasing the number of free carriers and Hall mobility. Both the carrier concentration and Hall mobility rapidly drop when the tin content rises from 5 to 7 atomic %. The 5 at.% Sn-doped TiO_2_ NTs exhibit noticeably increased electrochemical performance and rate capability, according to XPS measurements. The continuous redox behavior of TiO_2_ before and after Sn doping corroborates the results of XRD, Raman, and XPS and demonstrates that appropriate doping does not alter the anatase structure of pure TiO_2_. According to SEM analyses, the diameter is around 10 nm, and the average length is 150 nm. After tin doping, the tubes’ diameter and length remain unaltered. In contrast, and in line with SEM, the NTs are more evenly distributed following tin doping.	[92]
22	Anodization was conducted under various conditions	Amorphous, TiO_2_ NTs having a length of 950 nm and an inner diameter of roughly 60 nm. Nb Anatase Length 1 μm to 18.1 μm. Diameter 120 to 97 nm (the tube diameters were obtained by calculating the average values from ten NTs) Nb	Ti-Nb-O NTs, which demonstrated good bioactivity for mesenchymal stem cell adhesion, proliferation, and extracellular matrix formation, were obtained by anodizing the surface of a Ti35Nb alloy. Amorphous Ti-Nb-O NTs showed much higher in vitro bioactivity when compared to undoped TiO_2_ NTs and porous Ti-Nb-O without an NT structure. It is possible to add transition metal dopants to TiO_2_ NT arrays by anodizing titanium/transition metal alloys. To avoid the selective dissolving brought about by the heterogeneous distributions of components in multiphase alloy sheets, anodizing Ti-20Nb alloy sheets favored an alloy with a single-phase microstructure for the production of homogenous NT arrays. Another method for creating reliable NT arrays involved anodizing the two-phase Ti-20Nb-FC alloy in electrolytes with a low dissolving power. According to the results of the morphology analysis, the microstructure of the titanium/niobium alloy and the electrolyte’s dissolution force are both essential components in the creation of the NT structure.	[93,94]
23	TiO_2_-NTs synthesized by a hydrothermal method. Copper ions were assembled onto the surface of TiO_2_-NTs via reactions between Cu(NH_2_CH_2_CH_2_NH_2_)_2_(OH)_2_ and hydroxide radicals on the surface of TiO_2_ NT	Anatase, TiO_2_-NTs with uniform diameters around 10 nm and lengths around several hundreds. Cu^2+^	Rhodamine B is degraded more efficiently by copper ion surface-doped TiO_2_ NTs with good tubular architectures than by pure TiO_2_ NTs. The photoefficiency is raised by the efficient electron scavenger role of copper ions in preventing the recombination of photoexcited electrons and holes. Other transition metals can also be prepared using the same method to create modified TiO_2_ NTs using a surface-reaction-based assembly procedure.	[42]
24	Hydrothermal method	Amorphous, the average inner diameter of TiO_2_ NTs was 4 nm, while the average outer diameter was 10 nm, and their length was roughly 200 nm. They were hollow and open-ended. CeO_2_.	By inducing a red shift, moving the band gap to the visible light range, and successfully inhibiting the recombination of photoinduced electron–hole pairs, CeO_2_ in TiO_2_ NTs increased the photocatalytic activity. These substances have the potential to be employed as photocatalysts for the degradation of organic molecules in aqueous solutions because they are stable. Compared to P25, which had degradation efficiency of only 68% after 120 min of exposure, MB had photodegradation efficiency of 94.6% after 60 min and nearly 100% after 120. The capacity of TiO_2_ NTs to deteriorate was successfully improved by the composite of TiO_2_ and CeO_2_. When CeO_2_ was used as a dopant for TiO_2_ NTs, it was possible to see aggregated CeO_2_ nanoparticles on the surface of the TiO_2_ NTs that ranged in size from 5 to 10 nm.	[95]
25	Titania NT photoanodes were synthesized by anodization method.	Anatase, ≈120 nm outside diameter, Cr, Co, Cu, Fe, Mn	According to the findings of XRD, TEM, and SAED, doping with 3d TM ions has no effect on the crystal phase of TiO_2_ NTs. Without forming new phase structures, 3d TM ions were successfully doped into TiO_2_ NTs, as demonstrated by XPS and energy dispersive spectroscopy. The inclusion of 3d TM ions considerably improved the PEC water-splitting performance of TiO_2_ NTs, and UV–Vis absorption and visible light absorption both showed increasing absorption. M-TiO_2_ has photon energy and band gap energy of 3.09, 2.83, 2.75, 2.78, and 2.91 eV (M = Mn, Fe, Cu, Co, and Cr, respectively). Compared to the undoped NTs, the doped NTs’ photoluminescence (PL) intensity drastically dropped, demonstrating successful electron–hole pair separation. Fe^3+^, Cr^3+^, Co^2+^, Cu^2+^, and Mn^2+^ doped samples improve visible light absorption. Red absorption band is first shifted to 550 nm by doping with Cr^3+^, and then to 500–650, 550–700, and 500–700 nm by doping with Fe^3+^, Mn^2+^, and Co^2+^, respectively. TiO_2_ absorbs more in the ultraviolet region when Cu^2+^ is added. However, it also exhibits a sharp shift in the visible light spectrum between 600 and 800 nm and a significant absorption coefficient. Through “low-temperature doping” of TiO_2_ NTs, under visible light, 3d TM ions help TiO_2_ NTs to perform better at PEC water splitting.	[96]
26	Electrochemical anodization, the anodizing temperature was 25 °C, 35 °C, and 45 °C with anodization time of 5 min, 10 min, 20 min, and 30 min	Anatase and rutile, length 300 nm, NiO and Ni_2_O_3_	The thin-film X-ray diffraction pattern (TF-XRD) confirms that the nanotubular structure of the sample annealed at 450 °C is preserved, whereas it is partially collapsed in the sample annealed at 600 °C. Only the diffraction peaks of the matrix indicating the amorphous structure are seen in the TF-XRD patterns of the produced sample and the NT film annealed at 450 °C. In contrast, the typical peak of rutile TiO_2_ emerged at 27.5 °C after the sample was annealed at °C. After annealing at 450 °C for 1 h, no anatase TiO_2_ was found in the film NT, which may be easily obtained from the TiO_2_ NT film formed on pure Ti under the same annealing circumstances. At 600 °C, the TiO_2_ NT film grown on pure Ti undergoes a change from the anatase phase to the rutile phase, which might cause a partial collapse of the NT walls. The morphology, crystal structure, and surface chemistry of the Ni_2_O_3_-doped TiO_2_ NT film are affected by changes in the preparation processes, such as the anodization temperature, time, and annealing temperature. These changes in turn affect the NiTi alloy’s ability to resist corrosion, exhibit bioactivity, and be wettable. Rutile crystals were produced during annealing at temperatures of 600 °C, increasing the film’s bioactivity.	[97]
27	TiO_2_ NTs (TNTs) were synthesized by a hydrothermal approach. Co_3_O_4_ TNTs were synthesized by a wet chemical deposition precipitation method, also known as co-precipitation method	Anatase, diameters are approximately 10 nm Co_3_O_4_ (tricobalt tetraoxide)	The distinctive Co peak is visible in the Co_3_O_4_ TNTs’ XPS spectra at 780.1 eV, and the ratio of O to Ti grows noticeably. These findings show that the wet chemical deposition precipitation approach was successful in producing Co_3_O_4_ TNTs. TNTs feature multilayered walls and the usual morphology of NTs, according to transmission electron microscopy (TEM), and they have diameters of roughly 10 nm. The co-addition of Co_3_O_4_ TNTs increased the thermal stability of epoxy resin, according to a thermogravimetric study (EP). The cone calorimeter data showed that EP/Co_3_O_4_ TNTs had the lowest peak heat release rate and the lowest total heat release when compared to pure EP, both of which decreased by 35.4%. This shows that Co_3_O_4_ TNTs had an excellent synergistic flame-retardant effect and could significantly increase the flame retardancy of EP.	[98]
28	Electrochemical anodic oxidation of pure titanium, followed by annealing treatment	Anatase, diameter of these FeTiO_2_ tubes is in the range of 50–90 nm and their wall thickness is 50–80 nm. Fe	Due to its semi-full electronic structure and an ion radius near to that of Ti^4+^, iron doping of TiO_2_ NTs is a useful method for lowering the rate of electron–hole recombination and increasing the photocatalytic efficiency. In comparison to bare FeTiO_2_ NTs, Ag_2_S/FeTiO_2_ NTs have demonstrated higher photocatalytic activity in the degradation of MB. This may be because of the materials’ narrow band gap, sizable specific surface area, and effective heterojunctions.	[99]
29	Anodization, Ni-doped TiO_2_ NTs were fabricated by anodizing different Ti-Ni alloys.	Anatase and rutile, average diameter was 58.8 nm and the average NT length was 2.8 μm. Ni	The ability of undoped TiO_2_ to absorb light was enhanced by Ni doping. The separation of photo-generated electron–hole pairs was aided by moderate Ni doping. It was discovered that moderate Ni doping was a crucial design element for improved PEC properties. At 550 °C, Ti1NiO NTs demonstrated improved PEC water-splitting ability. The anatase to rutile phase transition benefited from Ni doping. Rutile modes became stronger with an increase in annealing temperature from 600 °C to 700 °C, according to XRD patterns, suggesting that an additional rutile phase could occur. Ti1NiO, which mostly comprised the anatase phase, was annealed at 500 °C. When the material was annealed at 550 °C, rutile phase peaks became visible.	[100]
30	TNTs were synthesized by a hydrothermal method. CeO_2_ TNT hybrids were prepared by wet chemical deposition precipitation method.	Anatase, NTs with multilayered walls and the diameter is around 10 nm. CeO_2_	CeO_2_ and TNT hybrid materials were successfully synthesized, as demonstrated by XRD, XPS, and TEM, and CeO_2_ particles were uniformly loaded on the surfaces of TNTs. Cone calorimeters were used to study samples’ combustion qualities and thermogravimetric analysis (TGA). The thermal stability and flame retardancy of matrix materials can be significantly improved by adding loaded CeO_2_ to TNTs, according to cone results. At 700 °C, the EP /0.1 CeO_2_ TNTs have the largest residual carbon content (19.8%). It also has the slowest rate of degradation; the PHRR and THR are 680 kW/m^2^ and 32.9 MJ/m^2^, respectively, reduced by 38.2% and 23.1% from the raw epoxy resin. TNT in EP creates a cross-linked network structure that can function as a physical barrier. The creation of the protected carbon layer and the efficient inhibition of the emission of both quality and quantity of heat from TNT loaded with CeO_2_ can produce a flame-retardant effect. CeO_2_-loaded TNTs can significantly increase the structural stability and extensibility of the carbon layer.	[101]
31	Chemical bath deposition (CBD) technique applied for the formation of CdSe/TiO_2_ NTs	Anatase, diameter of 50 nm, CdSe	A maximum photocurrent density of 0.0016 A/cm_2_ was displayed by CdSe/TiO_2_ NTs after 30 min of immersion in a CdSe precursor solution, an improvement of around 10% over pristine NTs when exposed to solar light. When CdSe species were not loaded onto the TNTs in amounts greater than 0.5 at%, the characteristics of the potential photocurrent density could be seen. These might work well as electron scavengers to reduce recombination losses when exposed to solar radiation. A saturation level may be attained by overloading the CdSe-containing TNTs by 0.5%. The creation of CdSe spheres near the opening of the NT arrays could have adverse effects because of the significant number of charge carrier recombination locations.	[102]
32	Alkaline hydrothermal process in a single step without additional calcination and reduction treatment	Rutile and anatase, length 100–200 nm, pore diameter 9.5 to 11.5 nm. Ag	The Brunauer–Emmett–Teller (BET) surface areas of the unmodified TNT and Ag_2_0 TNT samples were 392 and 330 m^2^g^−1^, respectively. TEM images showed that the TNT sample had 100-200 nm long openings on both sides. The adsorption edge of the Ag-modified sample TNT clearly shifted to the blue region. OH groups, pre-adsorbed H_2_O, and oxygen vacancies were found on the surfaces of the resultant samples, according to FTIR, PL, and XPS investigations. By assessing the removal effectiveness for Hg0, MG, CV, or the combination of both dyes under UV irradiation, the photocatalytic activities of the resulting samples were examined. The more AgCl and Ag species that were present on the surface TNT treated with Ag, the longer it took for the e^−^/h^+^ couple to separate. For Hg0, CV, MG, and the mixture of dyes, the Ag_2_0-TNT sample had better removal efficiency than the unaltered TNT sample. Successfully used for photochemical degradation of carcinogenic dyes and Hg0 in the gas phase.	[103]
33	Electrospinning Li_4_Ti_5_O_12_	Outer diameter of approximately 120 to 200 nm and their wall thickness is in the range of 30 to 40 nm. Li	With an initial capacity of 186.4 mA h g^−1^ and a capacity that remains constant after 100 cycles at a current density of 100 mA g^−1^, the Li_4_Ti_5_O_12_ NT/graphene composite specimen has the highest discharge specific capacities and cycle stability.	[104]
34	Sol–gel process, hydrothermal process, subsequent annealing in N2 atmosphere	Anatase, inner diameter of around 5 nm and an outer diameter of around 15 nm, pore diameter 9.37 nm. Sn	As compared to TiO_2_, Sn-TiO_2_, and Vo-TiO_2_, Vo-Sn-TiO_2_ has greater photocatalytic performance. With oxygen vacancies and Sn doping present in the TiO_2_ NT, the photocatalytic performance was improved. This is attributed to increased light absorption, increased specific surface area, and decreased electron–hole pair recombination.	[48]

## Data Availability

Not applicable.

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
