# Peer review of "Experimental Studies on TiO2 NT with Metal Dopants through Co-Precipitation, Sol–Gel, Hydrothermal Scheme and Corresponding Computational Molecular Evaluations"

_materials, 2023, doi:10.3390/ma16083076_

Round 1

Reviewer 1 Report

The authors of the review “Experimental studies on co-precipitation, solgel, hydrothermal scheme of TiO2 NT with metal dopants and corresponding computational molecular evaluations” made a systematic literature analysis. They made a review of different methods of synthesis of metal-doped TiO2 nanotubes and have shown the area for its application as the photocatalyst, photoelectrochemical water splitting, and photovoltaics. Much attention is paid to methods of alloying and doping titanium dioxide with atoms and metal ions, which makes it possible to change the magnetic properties of materials, as well as their electronic structure.

There are many typos:

1.     Figures (1-3 and 5-10) are cropped, maybe due to wrong formatting.

2.     Line 42 physicochemical

3.     Line 60 ions

4.     Line 129

5.     Line 135 - the comma instead of the dot

6.     Line 174

7.     Line 193 subscripts

8.     Line 299

9.     Line 555

I recommend to accept this manuscript after minor revision.

Author Response

Reply

  1. The authors of the review “Experimental studies on co-precipitation, solgel, hydrothermal scheme of TiO2NT with metal dopants and corresponding computational molecular evaluations” made a systematic literature analysis. They made a review of different methods of synthesis of metal-doped TiO2 nanotubes and have shown the area for its application as the photocatalyst, photoelectrochemical water splitting, and photovoltaics. Much attention is paid to methods of alloying and doping titanium dioxide with atoms and metal ions, which makes it possible to change the magnetic properties of materials, as well as their electronic structure.

Response:

Thank you for your thoughts, reflections and comments.

  1. There are many typos: Figures (1-3 and 5-10) are cropped, maybe due to wrong formatting.

Response:

Thank you for your comment, it was due to the conversion of our Word document to a PDF document. To overcome this difficulty and the resolution of the images, we have produced the revised manuscript in Tex format and now Figures 1-3 and 5-10 do not appear to be cropped. All figures were rechecked for resolution before being included in the revised version.

  1. There are many typos: Line 42 physicochemical, Line 60 ions, Line 129, Line 135 - the comma instead of the dot, Line 193 subscripts, Line 299, Line 555

Response:

Thank you for your comment. These typographical errors have been corrected in the revised manuscript and are marked with green text

We believe that the corrections have improved the quality of our work for which we are grateful.

Sincerely,

Dr. Saravana Prakash Thirumuruganandham

(https://orcid.org/0000-0003-4210-1363)

Reviewer 2 Report

1. Review all graphics, subtitles are small, ariel, no pattern. This seems irrelevant but it organizes the work for the reader.

2. The quality of the figures is poor, it should be improved.

3. “The template on which the particles are deposited, the content of the colloidal solution, the temperature, and the period of deposition all affect the particle size and the dimensions of the nanostructures.’ this part should be updated some refs, such as ACS Appl. Mater. Interfaces., 2021, 13, 12463−12471; J. Alloy. Compd, 2022, 897, 163178; Catal. Sci. Technol., 11(2021) 3946–3989 and Mater. Today. Commum., 2022, 31,103514.

4. “Characterization and its applications (Observation)” this part is very complicated, please simplify it carefully.

5. There is no full photocatalytic mechanism diagram

Author Response

attached pdf 

Reviewer 3 Report

The review should cover current trends and topical topics. There is no discussion of why a review of the present subject is necessary and who will benefit from it. The manuscript lacked review structure and was unrelated to the title or manuscript structure. Each part lacked a paragraph, making it difficult to read and comprehend. The conclusion should have been explored with future scope, since it seems to be an introduction and is a bit long. The authors did not take the manuscript seriously since there is no organization and the pictures are not clear, shading, and half of the image is missing in several figures. The table needed a title. In general, there should be a schematic diagram in the introduction describing the structure of the review to help readers understand the topic, but it is also lacking. Therefore, I reject the manuscript since the review structure/organization/subject may not be of interest to readers.

Author Response

Ambato, Ecuador, 28/02/2023

Dear Reviewer, Materials, MDPI

Kind greetings

Thank you for considering our manuscript entitled " Experimental studies on co-precipitation, solgel, hydrothermal scheme of TiO2 NT with metal dopants and corresponding computational molecular evaluations."

We are very grateful to you for giving us enough time to submit the revised manuscript. We have used this time to review the classical, quantum chemical calculations, experimental analysis related to Tio2 Nanotube- and relevant list of metal doping agents. In addition, we have subjected the manuscript to English proofreading and corrected the grammatical and structural form of the same.

All changes are colored green in the revised manuscript. In addition, blue font is a new addition which indicates the answer to the reviewer's questions.

Reply

The review should cover current trends and topical topics. There is no discussion of why a review of the present subject is necessary and who will benefit from it. The manuscript lacked review structure and was unrelated to the title or manuscript structure. Each part lacked a paragraph, making it difficult to read and comprehend. The conclusion should have been explored with future scope, since it seems to be an introduction and is a bit long. The authors did not take the manuscript seriously since there is no organization and the pictures are not clear, shading, and half of the image is missing in several figures. The table needed a title. In general, there should be a schematic diagram in the introduction describing the structure of the review to help readers understand the topic, but it is also lacking. Therefore, I reject the manuscript since the review structure/organization/subject may not be of interest to readers.

Response:

Thank you for your comment, We appreciate and value the observations, the manuscript has a comparative table scheme, the purpose of this work is essentially to investigate in depth the studies carried out in the experimental and computational field on TiO2 nanotubes. We explain our views for the following of your observations.

  1. The manuscript lacked review structure and was unrelated to the title or manuscript structure

Response:

We have introduced the introduction in the revised manuscript with the following text, which can be seen as blue text:

“Due to its many beneficial characteristics, titanium dioxide is a highly valuable substance that is used in a wide range of industrial products, including electronics, photo voltaics, paints, and food-grade materials. The formation of nanotubes in TiO2 phases and their interactions with other metal dopants, as well as biomolecule-polymer interactions and interfacial reactions are generating promising results in the field of renewable energy and medicine. However, much remains to be discovered about the properties of TiO2 nanotubes, and considering that there are numerous TiO2”

For the tittle of the manuscript : “Experimental studies on co-precipitation, solgel, hydrothermal scheme of TiO2 NT with metal dopants and corresponding
computational molecular evaluations” and this can be seen in final part of the introduction section with the following text

We have presented the DFT computational analysis for the doping of TiO2 nanoparticles, consequently the experimental synthesis methods of TiO2  NTs with metal dopants, the recent trends in the computational costs for the calculation of NT by classical and quantum molecular dynamics potentials, and in the last section, we detailed the future prospects of the synthesis methods for metal doping of various TiO2  nanostructures such as nanobelts and nanohelix of NT, which are important for renewable energy and other potential biological significance.”

And as for the structure and outline of the manuscript, this can be reflected with the following sections, in which relevant information has been included and discussed.

  1. DFT for materials, 3.Quantum computational verification of experimental doping of metals on TiO2 nanoparticles, 4. Experiments on doping of metals on TiO2 NTs, 5. Current demands to understand the TiO2-NT – metal doping through Classical and Quantum molecular dynamics (MD) to support experimental synthesis and
    characterization, 6. Conclusion

  1. Each part lacked a paragraph, making it difficult to read and comprehend

Response:  Thank you very much for your observations, In the revised manuscript we have corrected many of the sections, for example, in particular, we have added the following text, highlighted in blue, to section 5. It is carefully considered in the design and flow of the review manuscript

“On the other hand, important studies on the mechanism of dynamic surface  electron transfer have been carried out using Pt/Rutile TiO2 of 3 nm size and simulation cells at X, Y and Z = 6.43 nm and 31.78 nm as a type of effective photocatalysts that allow to demonstrate significant advantages [176 ], despite the fact that there is no classical MD has recently been made for Pt-doped TiO2 NT. The extremely small PT nanoparticle with a diameter of 3 nm that deposited on the TiO2 surface could be examined thanks to MD  simulation in combination with a reactive force field (REAXFF). Their MD results confirm  the experimentally observed method for regulated protection of highly active sites (such as  PT, AU, and PD) on supports, which is essential for the development of efficient and reliable photocatalysts. The force field of optimal atom potentials for liquid simulations was used   fixing the TiO2 substrate preserved the stable structure [ 177 ]. In the same way, simulations  were carried out to investigate the mechanisms of wetting and regulation, specifically, how  does the variation in the Li+ concentration affect the interfacial structure between ILS and TiO2?. What effect does this have on the contact length and angle of the electrolyte droplets on the TiO2-B (1 0 0 0) surface?. Doping with Li+ is predicted to slow down the dynamic wetting process and will alter the final shape of the ionic liquid droplet. For this purpose,  optimized condensed phase molecular potentials were used for MD (compass) studies that allowed to investigate the structures and dynamic properties of a 1 M KOH solution on Anatase TiO2 (001), (100) and (101) surfaces to understand the molecular detail of the effect  of KOH on the water splitting of the PEC phenomenon [178]. This study confirms that K+ ions prefer to bind to O2C on the TiO2 surface. Furthermore, the layering of oxygen  and water atoms on TiO2 surfaces significantly regulates the kinetics of potassium ions.  

The first study of substantial amounts of water behavior on the surfaces of anatase(101) nanotubes with a diameter of 1 nm uses traditional MD simulations [179 ], and the results  show that water inside the tube diffuses more slowly than water in contact with the outer
surface tint. The anatase phase of TiO2 NTs would allow them to create nanoscale ion
beams and serve in place of the huge magnets currently used for beam steering, according
to studies that have recently focused on the relatively restricted channeling property of
TNT [ 180 ]. Based on the Lindhard planar channeling potential, (MD) simulations were
used to examine the channeling of HE++ mega-electron ions (MEV) in titania nanotubes.
The ion channeling phenomenon made it possible for titania nanotubes to have a diameter of 100 nm and 2 m and a length of 1 m and 2 m, as shown in the simulated trajectories
of ions projected onto them. To study the gas-sensing properties of TiO2, Natalia et
al [181 ] used all MD atoms to model a 150 x 150 surface area of atomic planes of 100
anatase TiO2 and TiO2:MoO3compounds to study their interaction with water, hydrogen,
methane and methane and methane and ethanol molecules at temperatures of 300 K and
573 K. The synthesis of anatase-modified TiO2 and TiO2: MoO3 composite materials with
variable MoO3content by the SOL-gel method to verify the results of the simulations of
molecular dynamics. On the basis of the synthesized oxide materials, single-electrode
thermocatalytic chemical gas sensors were constructed, experimental tests were carried
out on their detection abilities of hydrogen, methane and ethanol gases. According to IR
spectroscopy data, the number of surface OH groups and adsorbed water molecules in
TiO2:TiO2 composite is significantly higher than in bare TiO2, and decreases with increasing MoO3 content. This experimental finding is reliable with the results of the (MD) simulation on the adsorption of the water molecule on the TiO2 and TiO2 surfaces. To explore the formation of non-abundant TiO2 crystalline thin films (both Rutile and Anatase),

Houska et al. [ 182 ] pulverized with a magnetron, atomic-level MD simulations with a Buckingham  interaction potential, using a simulation technique that is iterative.

 They estimated the  effect of the energy of the arriving atom, the structure of the substrate and the lateral size  (growing crystal) and the temperature of the substrate, and confirmed that the phenomena observed experimentally (in compared to anatase, the growth of rutile at higher energies per atom of the film and a higher temperature of the nucleation support temperature) are consistent with the MD results.

The results shed light on the intricate relationships that
exist between process parameters and the structures of the TiO2 films that are deposited.
Using high-resolution transmission electron microscopy (TEM), in situ ion irradiation TEM, and (MD) simulations, amorphous TiO2 nanotubes were compared with their crystalline counterparts, anatase TiO2 nanotubes. According to the calculations of MD [183 ] , the  internal stresses caused by the densification process during crystallization cause partially crystalline tubes to bend. TiO2 nanotube-enhanced biomedical devices have been shown to significantly affect mesenchymal stem cell proliferation, differentiation, and adhesion.
Cells react to the nanotubes by increasing adhesion, proliferation, and differentiation. In a
biological environment, proteins act as an intermediate layer on the surface of the material  to further promote cell adhesion and proliferation, and protein adhesion is the first event to occur when the implant surface makes contact with the biological environment. (tissue, body fluids). Smaller diameter TiO2 NTs have a greater surface area for the binding of positively charged proteins than larger diameter NTs [ 184 ] . Histone has the most adsorbed rotein in 100 nm diameter nanotubes (10 nm in length), while 15 nm diameter nanotubes have higher values

And also, we indicate that, to date, numerous material developments have been initiated for doping TiO2 with metals, and we have emphasized the geometry of nanotubes over other forms such as nanoparticles and nanowires. Therefore, the recent history of each metal and its doping with TiO2 nanotubes is important to this review manuscript. Moreover, each individual metal has its own unique properties as shown by many experiments. For example, the effect of platinum islands in uniformly doped Au has been studied at TNT compared to non-uniformly doped Au, and Au as a dopant metal is completely different compared to other dopants. We have created a table to characterize the nature of the metal of interest so that it can be easily identified.

  1. The conclusion should have been explored with future scope, since it seems to be an introduction and is a bit long

Response:

Many thanks for the suggestion. In the revised manuscript, we have included the following text in the conclusion section as a blue text to illustrate the future scope and associated importance of using new classical computational methods such as Reachtive forcéfield to study TiO2 nanotubes.

In the same way, another relevant aspect is that the classical (MD) simulations that are
currently available and are experimentally challenging include the formation of nanotubes
in the aforementioned TiO2 phases and their interactions with other metal dopants, as well
as the interactions between biomolecules and polymers and interfacial. We believe that
experiments with various doping elements, like DFT research, similar processes can be
replicated using classical MD. Furthermore, a modern technique for simulating Ni-doped
MoS2 to understand the effect of doping was found to be known as reactive force field
(ReaxFF), connecting quantum mechanical and experiments [ 186 ]. This makes TNT metal doping using ReaxFF a particularly good way to understand the dynamics of doping
research [187].”

Since we deal with both the computational and experimental aspects, the section on conclusions is somewhat extensive.

  1. The authors did not take the manuscript seriously since there is no organization and the pictures are not clear, shading, and half of the image is missing in several figures. The table needed a title. In general, there should be a schematic diagram in the introduction describing the structure of the review to help readers understand the topic, but it is also lacking.

      Response:

Thank you for the comments,  it was due to the conversion of our Word document to a PDF document., Now, in the the revised manuscript “ Tex” format  is adopted and now Figures 1-3 and 5-10 do not appear to be cropped. All figures were rechecked for resolution before being included in the revised version.

We have given the tittle for the table  this can be seen as a blue text in the revised mansucript “ Different methods of Experimental synthesis, Characterization and its applications of metal doping  for TiO2 nanotubes”

Instead of the schematic diagram, we created a new figure that could explain the meaning of the photocatalytic degradation mechanism of metals doped with Tio2. This figure can be referred to as Figure 1, whose caption is highlighted with blue text “Mechanism of TiO2 photocatalysis: hv1 : pure TiO2 ; hv2 : metal-doped TiO2 and hv3 :nonmetal-doped TiO2 .”  This figure is included in the introductory section of the revised manuscript

We believe that the corrections have improved the quality of our work for which we are grateful.

Sincerely,

Dr. Saravana Prakash Thirumuruganandham

(https://orcid.org/0000-0003-4210-1363)

Reviewer 4 Report

The present manuscript reviews TiO2 nanotubes modified by metals for versatile applications. TiO2 is one of the most studied oxides for catalytic and energy applications. Therefore, there are already some review papers regarding its fundamental properties as well as applications in different directions. The present manuscript mainly focuses on nanotubes and metal-induced modifications. I first suggest the authors add comments to distinguish this review from other similar reviews. 

Another important but, unfortunately, the missing point is distinguishing the TiO2 nanotubes from other forms of TiO2 nanostructures. 

Some figures, such as Fig 1, Fig 3, Fig 6, etc., have poor representation. Unfortunately, half of these figures are not even visible to the reader. This makes the present version nonreadable and non-reviewable. 

The table on pages 14-25 contains the redundantly long text. It is difficult to follow what is the essence of this table. Also, its caption and numbering are missing.

The abstract and conclusion don't present the essence of the reviewed subject. 

I suggest the authors carefully investigate relevant papers since the current reference list doesn't seem comprehensive.

There are many papers to review, especially regarding the difference between the TiO2 NTs and other nanostructures and their combination with other materials.

https://doi.org/10.1002/aoc.5005 ; https://doi.org/10.1021/acscatal.0c00556 ; https://doi.org/10.3390/catal11070857

I suggest the authors consider all these references and also investigate the recently published TiO2-related literature more carefully for extensive discussions. 

In general, the manuscript contains severe errors, typos, and important comparisons of literature are missing from the discussions. The figures and table need to be optimized. In the present form (especially with hidden figures, it is difficult to read and review the manuscript. All these must be corrected for the clear review process. 

Round 2

Reviewer 2 Report

accept

Author Response

Dear Reviewer

Thank you very much for your review report and approval of our manuscript. We kindly inform you that, We have made a few changes to the content of the work, added two figures, and modified the title of the manuscript in its revised version.

Thank you & Sincerely

Saravana

Dr. Saravana Prakash Thirumuruganandham
(https://orcid.org/0000-0003-4210-1363)

Reviewer 3 Report

The authors did not make any modifications to my recommended points. Figure 1 does not expose the subject and content of the present paper. Figure 4 is missing in the manuscript. There are yet no sub divisions or thematic material. There are several typographical errors and missing figure descriptions in the text. The title of the manuscript is so unclear as to whether it is a research or review article. It should be straightforward and simple for the reader to understand the manuscript. Abstract is unclear and missing in some details. The current form of the text may not interest to readers. I thus cannot accept the manuscript.

Reviewer 4 Report

The manuscript can be accepted with no further corrections.

Author Response

Dear Reviewer

Thank you very much for your review report and approval of our manuscript. we kindly inform you that we have made a few changes to the content of the work, added two figures, and modified the title of the text in its revised version.

Thank you & Sincerely 

Dr. Saravana Prakash Thirumuruganandham
(https://orcid.org/0000-0003-4210-1363)